# Drone-based medication delivery for rural, flood-prone coastal communities

**Yin-Hsuen Chen**[1]*, **Amro M. El-Adle**[2], **Kevin J. O'Brien**[3],
**Taylor Wentworth**[4], **Heather G. Richter**[5,6]

**1** Center for Geospatial Science, Education, and Analytics, Division of Digital Transformation & Technology, Old Dominion University, **2** Department of Information Technology and Decision Sciences, Strome College of Business, Old Dominion University, **3** Virginia Modeling Analysis & Simulation Center, Office of Enterprise Research and Innovation, Old Dominion University, **4** Department of Mathematics and Statistics, Old Dominion University, **5** Joint School of Public Health, Mason and Joan Brock Virginia Health Sciences at Old Dominion University, **6** Old Dominion University-Thomas Jefferson National Accelerator Facility Joint Institute on Advanced Computing for Environmental Studies

* y3chen@odu.edu

## Abstract

Access to healthcare remains a critical challenge for rural populations, particularly in flood-prone coastal communities where transportation barriers limit access to essential medical services. This study evaluates the effectiveness of drone-based medication delivery in improving healthcare accessibility for vulnerable populations on Virginia's Eastern Shore. Compared to traditional personal vehicle travel, drone delivery reduced trip times from up to 50 minutes to under 10 minutes for more than 80% of the population, including elderly patients. Using publicly available datasets, we developed two transportation vulnerability indices that incorporate age, travel time, and flood risk to prioritize patients for drone-based pharmaceutical delivery. These indices were examined using Getis-Ord Gi* spatial analysis, which identified statistically significant clusters of high-need patients, particularly around the north-ernmost drone station. The results reveal that elderly residents in remote, low-lying areas are especially vulnerable to missed prescriptions due to both transportation barriers and flooding. Our approach demonstrates how drone delivery can reduce healthcare access disparities while offering a scalable and resilient framework for other medically underserved regions, especially under time or resource constraints.

## 1. Introduction

Access to healthcare remains a significant challenge for remote and underserved coastal communities, particularly in flood-prone regions where transportation disruptions can delay or prevent the delivery of essential medications. This barrier disproportionately affects vulnerable populations, exacerbating health disparities and limiting healthcare accessibility. Travel by traditional personal vehicles to retrieve

**Data availability statement:** All the geospatial outcomes are available from the Harvard Dataverse (webpage: https://dataverse.harvard.edu/dataverse/DroneDOTSMART).

**Funding:** Y.-H.C., A.M., K.O., T.W., and H.R. received financial support from the United States Department of Transportation's Strengthening Mobility and Revolutionizing Transportation (SMART) program. The grant number is 69A3552341006-SMARTFY22N1P1G54. The funder did not play any role in the study design, data collection and analysis, decision to publish, or preparation of the manuscript.

**Competing interests:** The authors have declared that no competing interests exist.

medication may not meet the needs of these communities due to infrastructure limitations, geographic isolation [1], and the increasing frequency of extreme weather events [2]. Unmanned aerial vehicles (UAVs), or drones [3], present a promising alternative for overcoming these logistical challenges by providing rapid, reliable, and scalable medication delivery [4]. However, the potential of drone-based systems to enhance healthcare access, particularly for vulnerable coastal communities frequently affected by flooding due to high tides [5,6], and tropical cyclones [7] has yet to be fully investigated. Addressing this gap necessitates a comprehensive evaluation of drone delivery systems, including time savings, geographic coverage, and the identification of high-priority areas with the greatest need for intervention, particularly those that are vulnerability hot spots.

Healthcare access remains a major challenge for rural communities. In the rural U.S., where over 46 million people reside, approximately 80% are medically underserved, facing complex barriers to healthcare access [8]. Berenbrok, Tang [9] investigated household access to pharmacies across U.S. counties using ArcGIS Network Analyst and classified counties by Rural-Urban Continuum Codes (RUCC). Accessibility varied markedly: 58.6% of residents in large metropolitan areas lived within 1.6 kilometers of a pharmacy, compared with only 26.9% in rural areas. Worldwide, studies have consistently shown significant disparities in access to pharmaceutical services between urban and rural communities (e.g., Todd, Copeland [10], Law, Heard [11]). Compared to their urban counterparts, rural residents often experience lower incomes, reduced healthcare literacy, limited broadband access for telehealth services, and inadequate transportation options for healthcare-related trips [8]. Tharumia Jagadeesan and Wirtz [12] reviewed studies on pharmacy accessibility and found that 11 of these examined pharmacy density in relation to urban and rural populations. All of the reviewed studies consistently reported that urban populations have better access to pharmacies compared to rural populations. These findings highlight the persistent disparities in healthcare access between rural and urban populations, underscoring the urgent need for innovative solutions to improve pharmacy accessibility in underserved rural communities.

Reliable transportation is essential for accessing healthcare services, and its absence has been linked to missed or delayed medical appointments and increased healthcare costs [13,14]. Studies highlight that elderly rural populations are particularly vulnerable, as they face compound challenges such as limited public transportation, lack of a driver's license, and financial constraints [1,15]. For example, Shirgaokar, Dobbs [16] found that non-driving status, low income, poor health, and disability significantly restricted healthcare-related travel among rural elderly populations. Similarly, Ranković Plazinić and Jović [17], in a study of 346 elderly respondents in rural Serbia, found that mobility levels were generally low, but accessibility varied among settlements, with age and possession of a driver's license emerging as key factors influencing mobility. Personal vehicles are essential for mobility and access to critical services in rural communities, particularly for older adults. A national study of individuals aged 65–79 found that rural residents were 7% more likely than their urban and suburban peers to emphasize the importance of driving [18],

underscoring their reliance on personal transportation. In rural North Carolina, individuals with a driver's license made 2.29 times more healthcare visits, while those with access to family or friends for transportation had 1.58 times more visits [19], highlighting how transportation directly affects healthcare access. Yet, despite this heavy reliance, older drivers in rural areas face disproportionate barriers due to chronic health conditions, physical impairments, and age-related declines in driving ability [20,21]. Without viable transportation alternatives, these challenges severely limit rural seniors' access to care and independence.

The transportation challenges are particularly acute in flood-prone coastal communities, where geographic isolation, rising sea levels, and frequent flooding events further restrict access to essential healthcare services [7,22,23]. Transportation network disruptions caused by extreme weather events exacerbate these accessibility issues. For instance, Hierink, Rodrigues [7] found that Cyclones Idai and Kenneth in Mozambique reduced healthcare accessibility from 79% to 53% and from 82% to 72%, respectively. Similarly, Tomio, Sato [24] examined the impact of flooding in Kagoshima, Japan, and reported that up to 23% of evacuated patients with chronic conditions experienced medication interruptions, with those aged 75 and older being particularly affected. These findings highlight the critical need for alternative delivery methods to ensure continuity of care during and after flood events. Given the vulnerability of traditional transportation methods to flooding and extreme weather, alternative solutions are essential to maintaining healthcare access during disruptions.

Autonomous drone delivery systems have emerged as a promising option, offering a reliable method for delivering medical supplies to isolated populations [4,25]. In the U.S., several private companies have received regulatory approval to provide drone-based delivery services in select locations [26], demonstrating the potential for this technology to improve healthcare accessibility in vulnerable regions. A notable milestone occurred in 2015, when the first government-approved medical drone delivery was conducted in Wise, Virginia. This event demonstrated the feasibility of using drones to navigate rugged, rural terrain and highlighted their potential for providing faster, more dependable access to critical supplies in hard-to-reach communities [27]. Later, two U.S.-based drone companies, Volansi and Zipline, piloted vaccine and medication deliveries in North Carolina and Arkansas [28].

Studies have further highlighted drones' ability to transport medicine, vaccines, and emergency medical supplies, with key benefits such as minimizing infection risk through reduced human contact and improving emergency response times [29,30]. Additionally, Haidari, Brown [31] found that drone-based vaccine distribution increased availability while reducing costs compared to traditional land-based transport. Regarding delivery costs, an analysis by PricewaterhouseCoopers (PwC) estimates that each drone delivery ranges from $6 to $25 per trip, with costs decreasing as the number of drones overseen by a single pilot increases [32]. In practice, Walmart currently charges $19.99 per delivery in the U.S., while Manna customer in Ireland pay about $4 per delivery [33,34].

Although drones offer a promising solution for overcoming these logistical challenges by providing rapid, reliable, and scalable medication delivery [4], research on their routine use for non-emergency healthcare services remains limited. Existing studies have primarily focused on emergency scenarios, such as delivering automated external defibrillators [35] or distributing vaccines during pandemics [36], leaving a critical gap in understanding the broader, sustained impact of drone-based deliveries on healthcare access in rural and flood-prone regions. While several studies have compared delivery methods such as electric vehicles, trucks, and bicycles with drones (e.g., Garus, Christidis [37], Comi and Savchenko [38]), few have examined personal vehicle travel against drone delivery, a comparison that could yield critical insights for designing effective drone delivery strategies in rural, flood prone communities. Nationwide studies, such as Berenbrok, Tang [9], have analyzed pharmacy accessibility disparities across the U.S. using household-level network analysis with a 1% random sample. Similarly, Sharareh, Zheutlin [39] employed population-weighted census tract centroid to estimate driving times to community pharmacies at a national scale. While valuable, these analyses remain coarse in scale, highlighting the need for more comprehensive, locally focused studies to capture finer-grained variations in accessibility. Additionally, frameworks like Kim, Lim [25] lack direct comparisons between drones and traditional transportation methods, preventing a comprehensive evaluation of drone efficiency, geographic reach, and service reliability. Together, this

gap underscores the urgent need for a comprehensive investigation into the role of drones in delivering essential medical supplies to vulnerable rural populations.

This study evaluates the effectiveness of drone-based delivery systems in enhancing healthcare access for flood-prone coastal communities, with a focus on serving the most vulnerable populations. The primary research objective is to develop and assess a scalable framework that integrates drone-based delivery, spatial vulnerability analysis, and operational modeling to improve equitable healthcare access in disaster-prone rural regions. Using Virginia's Eastern Shore (hereafter ES) as a case study, this study compares traditional vehicle-based travel with drone-based delivery systems in flood-prone coastal communities and evaluates patient vulnerability. Travel times were estimated for both modes, and two vulnerability indices—one incorporating age and travel time (VAT) and another adding flood impacts (VATF)—were used to identify high-risk populations. Hotspot analysis guided prioritization of drone stations, demonstrating how drones can improve timely medication access for the most vulnerable residents. This study contributes to healthcare accessibility and disaster-resilient delivery systems in three ways: by quantifying the time savings of drone-based medication delivery over traditional vehicle travel, by developing spatial vulnerability indices to identify high-need patients, and by evaluating drone hub performance to guide effective placement and resource allocation. Together, these findings demonstrate the feasibility and strategic value of drone networks as a resilient, equitable alternative to vehicle-dependent systems, offering a scalable framework for application in similar regions worldwide.

In what follows, Section 2 describes the study area and methodology, Section 3 presents the results of our analysis, Section 4 discusses these findings and the limitations, and Section 5 concludes with directions for future work.

## 2. Methodology

### 2.1 Study site

The ES forms the southernmost part of the Delmarva Peninsula, separated from the mainland by the Chesapeake Bay (Fig 1). The eastern portion features a complex of lagoons and barrier islands along the Atlantic Ocean, while the central and western parts are well-drained, with elevations reaching up to 17 m above sea level [40]. The peninsula's north–south axis follows a prominent local ridge, where U.S. Route 13 connects various villages and towns. The ES consists of two counties, Accomack and Northampton, with the only direct connection to mainland Virginia being the 27 km Chesapeake Bay Bridge Tunnel. To the northwest, Tangier Island is an isolated community only accessible by boat or plane, where medical deliveries typically require two to three days. This geographic isolation compounds healthcare access across the ES, particularly for remote communities like Tangier Island. Additionally, the risk of flooding and sea-level rise [41] exacerbates existing barriers to healthcare services, which may be especially important during emergencies [42]. Only seven pharmacies serve patients across almost 2,608 km² of the ES (Fig 1b). There is one bus line along a single main road, which provides limited access to smaller rural roads [43].

The ES was selected as our case study due to its rural nature, high flood exposure, and significant transportation barriers to healthcare. The region is designated as both a medically underserved area (MUA) and a health professional shortage area (HPSA) by the federal government [44]. Additionally, the ES is classified as a historically disadvantaged community (HDC), having been marginalized by underinvestment and overburdened by multiple disadvantage indicators, including health, transportation, and environmental resilience [45]. A recent health assessment revealed that more than 50% of residents identified transportation as the primary barrier to healthcare access [46]. Households on the ES also face lower median incomes, higher poverty rates, and higher rates of households lacking a personal vehicle compared to state and national averages [47]. Table 1 provides a comparison of community rates for hypertension, lack of health insurance, and limited access to broadband internet. Given that hypertension is a major healthcare concern in the region, patients with higher prevalence of hypertension were selected for the pilot study [46]. Additionally, flooding was incorporated as a key transportation barrier to assess community resilience.

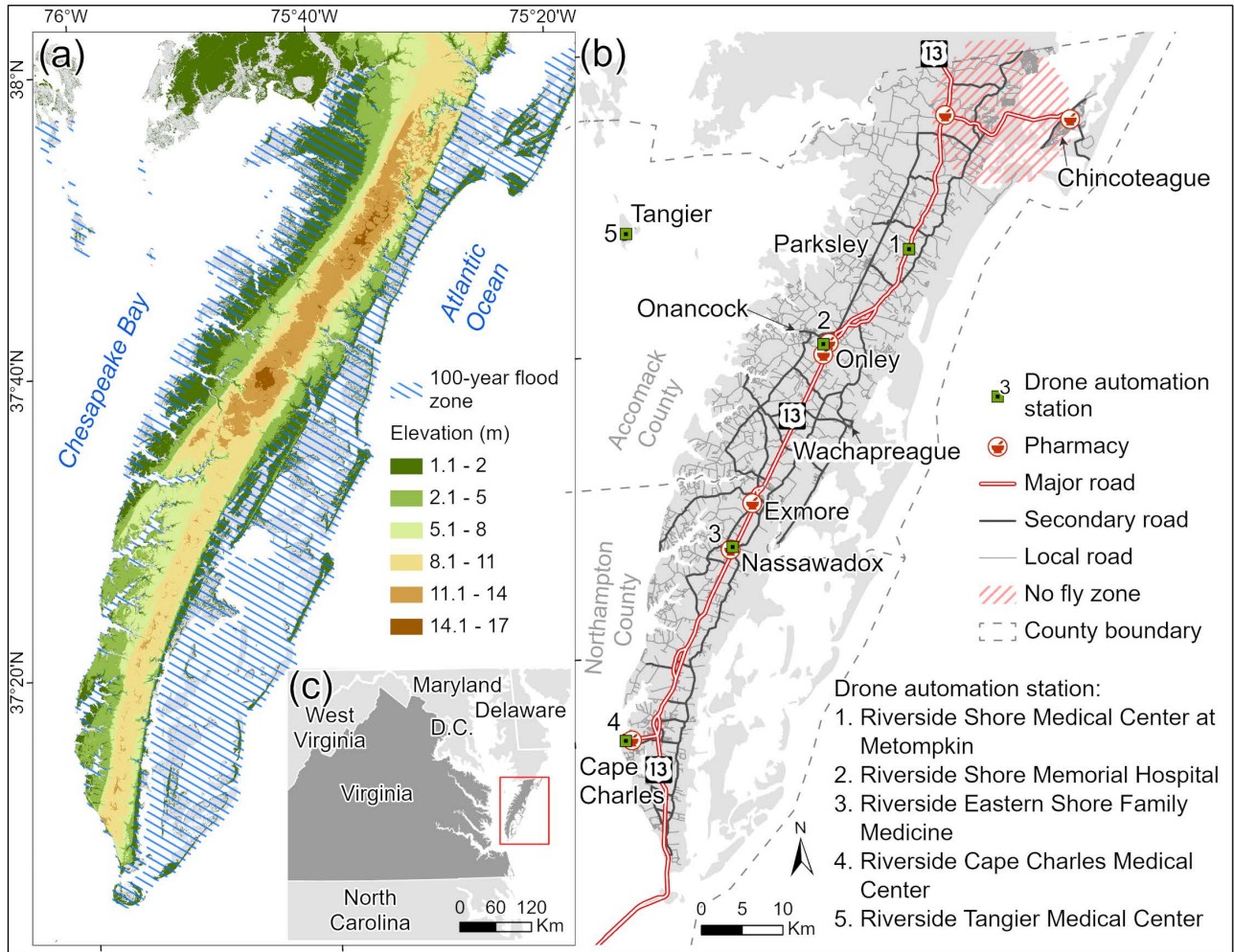

**Fig 1. A side-by-side map showing the 100-year flood zone along the elevation relief of Eastern Shore (ES) of Virgina (a), as well as the locations of drone stations, pharmacies, and the road network with elevation relief (b).** The inset map (c) highlights the study area's relative location on the Eastern Coast of the U.S., indicated by a red rectangle.

**Table 1. A comparison of socioeconomic indicators on the ES with state and national levels.**

| Indicator | Eastern Shore of Virginia | Virginia | United States |
|---|---|---|---|
| Median household income (USD) | $45,000 | $80,615 | $70,784 |
| Households living below the poverty line (Percent) | 18.0% | 10.6% | 11.5% |
| Persons 65 years and older (Percent) | 21.5% | 16.0% | 16.8% |
| Population density (persons per km$^2$) | 170.9 | 524.7 | 240.9 |
| Prevalence of hypertension (Percent) | 40.0% | 32.5% | 32.6% |
| High school graduate, persons aged 25 and older (Percent) | 80.0% | 89.0% | 88.5% |
| Persons aged 18 and older lacking health insurance (Percent) | 21.0% | 8.5% | 11.0% |
| Households without a vehicle (Percent) | 10.0% | 7.0% | 8.5% |
| Households with Broadband Internet Subscription (Percent) | 65.0% | 80.0% | 85.0% |

## 2.2 Data acquisition and pre-processing

Access to medical data is often limited due to privacy concerns, making secure data management costly [48,49]. In the U.S., healthcare data brokers charge from $0.05 to $125 per record, and entire electronic health record databases can cost up to $500,000 [50]. Alternatively, public datasets, such as those from the U.S. Census, offer free, standardized data. Studies like Kolak, Bhatt [51] have used public data to identify vulnerable populations based on social determinants of health, allowing for essential analysis without relying on medical records. To ensure the privacy of patient healthcare records, our analysis relies entirely on multi-sector public datasets. This approach not only safeguards patient health information but also renders the analytical tools amenable to diverse regions.

Geospatial data were obtained from various local, state, and federal agencies (Table 2) and processed to cover the ES, and ESRI® ArcGIS Pro 3.X was used to conduct the geospatial analyses. The coordinates of seven pharmacies and five drone automation stations on the ES were compiled for analysis. The pharmacies include Walgreens, CVS, Walmart, Rayfield Pharmacy Cape Charles, Rayfield Pharmacy Nassawadox, Atlantic Community Pharmacy Oak Hall, and H & H Pharmacy Chincoteague. The drone automation stations are Riverside Shore Memorial Hospital, Riverside Cape Charles Medical Center, Riverside Eastern Shore Family Medicine, Riverside Shore Medical Center at Metompkin, and Riverside Tangier Medical Center (Fig 1b). These stations are part of Riverside Health System, a leading healthcare provider on the ES.

To improve the accuracy of population segment computation, we utilized the Intelligent Dasymetric Mapping (IDM) Toolbox developed by the Environmental Protection Agency (EPA) [52,53]. We imported the IDM Toolbox [54] into ArcGIS Pro. The IDM tool requires input layers, including the census block, land use/land cover, and uninhabitable areas. The census block layer provides population data, which was used to calculate not only the total population but also the population of individuals aged 60 and older. This distinction is essential because the older population has a higher prevalence of hypertension: according to the 2017–2018 studies [55], hypertension affects 22.4% of adults aged 18–39, and 74.5% of those aged 60 and above.

**Table 2. Geospatial data utilized in analysis.**

| Data | Latest update[a] | Source |
|---|---|---|
| Road centerlines | June 2024 | Virginia Geographic Information Network (https://vgin.vdem.virginia.gov/pages/clearinghouse) |
| Parcel | July 2024; June 2024 | Accomack County GIS Data (https://accomack-county-virginia-open-data-portal-accomack.hub.arcgis.com/); Virginia Geographic Information Network (https://vgin.vdem.virginia.gov/pages/clearinghouse) |
| Building footprint | June 2024 | Virginia Geographic Information Network (https://vgin.vdem.virginia.gov/pages/clearinghouse) |
| UAS facility map data | July 2024 | Federal Aviation Administration (https://hub.arcgis.com/datasets/faa::faa-uas-facilitymap-data/about) |
| Decennial demographic and housing characteristics data | 2020 | Explore Census Data (https://data.census.gov/) |
| National land cover database | 2021 | Earth Resources Observation and Science Center (https://www.mrlc.gov/data/nlcd-land-cover-conus-all-years) |
| Flood insurance map | 2015 (Accomack County), 2016 (Northampton County) | Flood Map Service Center (https://msc.fema.gov/portal/advanceSearch) |

[a]Update dates are listed based on the acquisition date; sources may have more recent updates.

The land use/land cover layer served as an ancillary raster layer to determine the computational methods, which included preset density, sampling, or intelligent areal weighting [52]. We created a JSON file to specify which land cover classes were assigned a preset density of zero. These classes include land cover and land use for open water, barren land, deciduous forest, evergreen forest, mixed forest, shrub/scrub, grassland/herbaceous, pasture/hay, cultivated crops, woody wetlands, and emergent herbaceous wetlands, respectively. Additionally, an uninhabitable layer was created using the method developed by Baynes, Neale [54] to exclude specific areas from the dasymetric computation. These uninhabitable areas include government-owned lands, major road networks, and commercial land parcels. The output from the IDM toolbox is in raster format, where each pixel's value indicates the estimated population at that specific location. Incorporating these layers and computational methods enhanced the precision of population distribution estimates, particularly for older patients.

Building footprints were also used to identify potential patient sites. Due to the absence of zoning information, building footprint data from the uninhabitable layer was excluded [54]. The building footprints were overlaid with the parcel layer, and each building footprint was assigned a parcel identification number. Within each parcel, the largest building identified was assumed to be the primary residence of the patient. The results of the dasymetric mapping were used to perform zonal statistics, enabling the estimation of population within various travel zones. Additionally, building footprint data were used in the network analysis to identify precise locations of potential patients. This combination of methods ensured an accurate assessment of both population distribution and patient locations within the study area, facilitating better planning and resource management. Finally, due to its proximity to a small airfield, the Federal Aviation Administration (FAA) has designated this area of the ES a restricted zone for drones. As a result, residents of this area, totaling 7,749 individuals, including 2,943 aged 60 and older, are excluded from the analysis.

## 2.3  Operational and regulatory considerations of medication delivery by drone

Prior studies have suggested partnerships between healthcare professionals, transportation experts, and community liaisons to overcome complex transportation barriers [56,57]. This case study involves collaboration among local stakeholders, healthcare professionals, drone delivery providers, and academic researchers. This transdisciplinary approach addresses the complexity of transportation barriers and offers data-driven insights into optimizing drone delivery networks. The organizations partnering for this study are shown in Table 3 alongside their role in the project [58]. Given the novelty of drone delivery technology and the importance of delivering prescription medications safely, this section describes important regulatory, institutional, and operational considerations.

From a delivery hub, drones may fly to patient locations more than 6 miles away, thereby avoiding traffic, or the need to navigate rural roads that may be flooded. As shown in Fig 2, drone delivery allows patients to retrieve medications from their lawn or backyard, rather than needing to travel to a hospital or pharmacy via public transportation or personal vehicle. This autonomous delivery mechanism may be especially beneficial for patients managing multiple conditions with

**Table 3. Overview of partner organizations.**

| Organization | Role |
| --- | --- |
| Accomack-Northampton Planning District Commission | Community Engagement/Public Outreach |
| DroneUp | Technical and Service Provider |
| Old Dominion University/Virginia Institute for Spaceflight & Autonomy | Project Management/Data Modeling |
| Riverside Health | Medical Service Provider |
| Virginia Innovation Partnership Corporation | Funding Support |

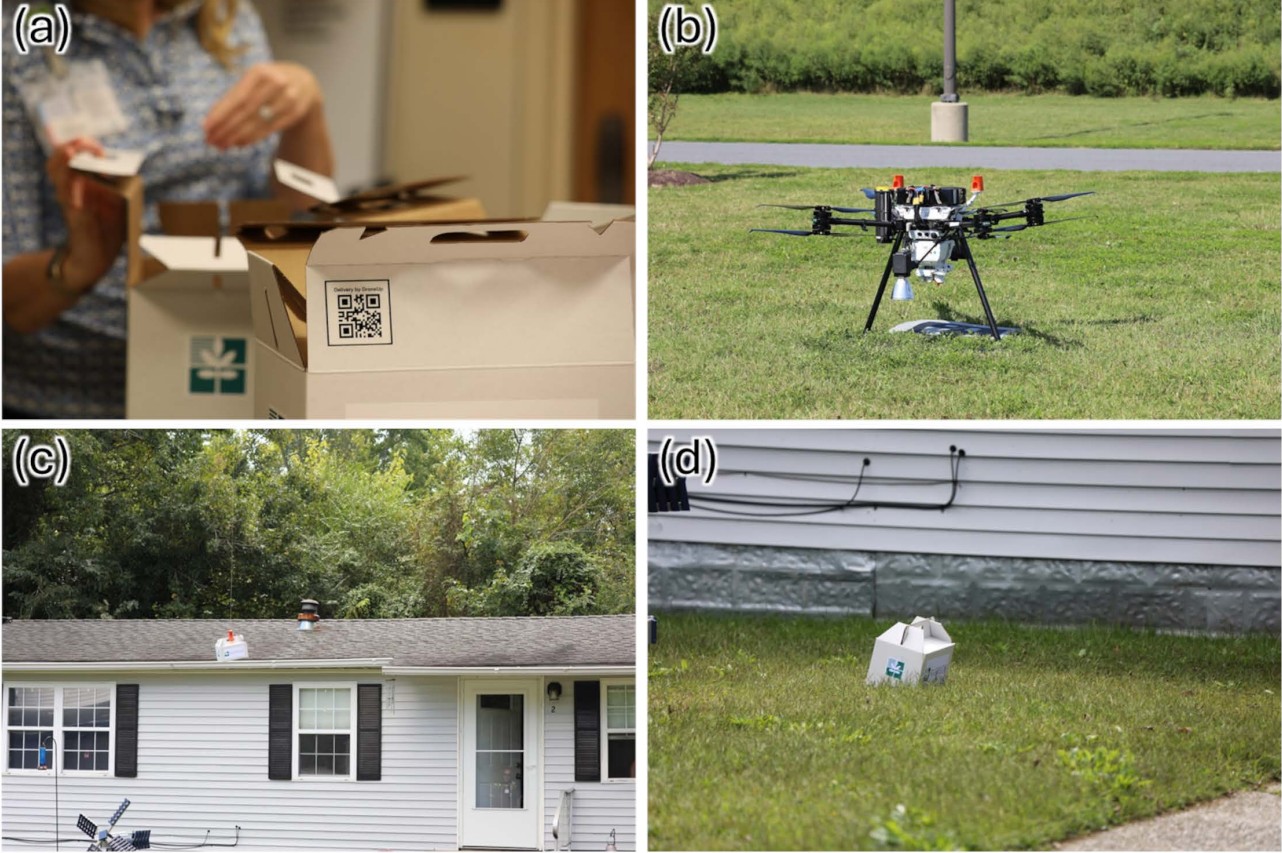

**Fig 2. Stages of drone delivery of hypertension medications to patients on the Eastern Shore of Virginia: (a) Medications are prepared and packaged by a healthcare professional; (b) the drone is loaded with the package and a charged battery; (c) the medication is lowered to the patient's residence using a winch line; (d) the package is left on the lawn for the patient to retrieve.** Photos Courtesy of Virginia Institute for Spaceflight & Autonomy as part of the Elevating Healthcare Access Project.

frequent medication refills, or patients who may not be able to drive. While drone delivery platforms require infrastructure investments, those fixed costs can be spread over a large volume of deliveries, reducing the cost per delivery.

Although the procedures outlined here were used on the ES for this specific case study, they may be generalized more broadly in similar communities. As shown in Fig 3, deliveries commence at a drone hub, at which qualified healthcare workers pack a prescription into a carton to be carried aboard the drone [58]. To meet state and federal regulations, DroneUp and Riverside Health use a validation system that ensures not only that the hypertension prescriptions are suitable for drone delivery, but also that the patient address is cross-validated prior to drone launch. While being loaded with a parcel, the drone's battery is also swapped with a freshly-charged unit to permit maximum flight time. For the earliest stage of the partnership described in this article, DroneUp received permission from the FAA. Each drone is piloted remotely by a single pilot. The drones are equipped with cellular communications equipment to permit remote piloting [58]. DroneUp also performs extensive scans of the flight paths in advance of delivery operations to create a digital twin of the environment to ensure that in case the drones lose connectivity, they can land safely in an emergency. Finally, drones are designed to launch from and to return to any hub in the DroneUp network, rather than being required to return to the hub from which the drone launched. This allows the drones to fly for a longer range as compared with a system in which each drone must return to the same hub from which it launched [58].

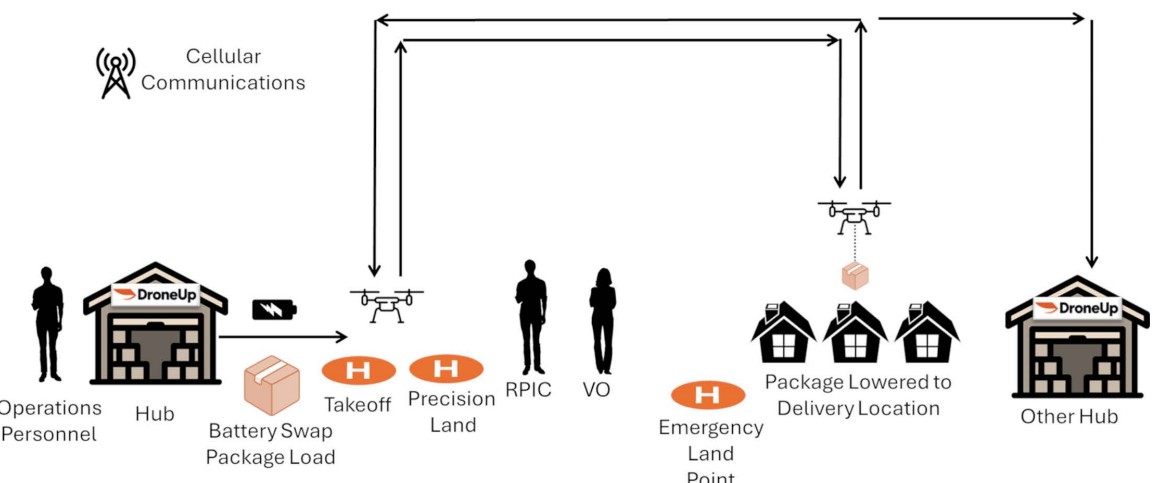

**Fig 3. Conceptual illustration of medication delivery by autonomous drones.** Adapted from Federal Aviation Administration [59].

For the flights described in this article, DroneUp deployed a visual observer to maintain line of sight of the drone at all times as a precaution. For future deployments, DroneUp has applied for FAA approval to fly drones beyond the visual line of sight, eliminating the need for a visual observer. The approval would also permit a single remote pilot to operate several drones simultaneously [58].

### 2.4 Comparison of travel zones

Given patient locations, this subsection describes tools that delineate travel time zones for pharmaceutical trips conducted via personal vehicle, as compared with drone-based deliveries. Personal vehicular trips consider round-trip travel times, since the patient would be required to drive to the pharmacy, then to drive home with the prescription medication. But for drone delivery, only one-way travel times from the nearest drone station to the patient's address are computed, since the medication is delivered upon the drone's arrival to the patient's address.

To estimate travel time for vehicular trips, we first constructed a Network Dataset, accounting for road hierarchy and travel speed (in minutes), the latter of which was determined by dividing the road length by the speed limit. Road networks were classified into three hierarchical levels: Level 1 included major US and Virginia highways; Level 2 included local main arteries; and Level 3 included local secondary arteries and others (Fig 1b). This hierarchy system assumes that drivers are more likely to utilize major roads with higher speed limits [60]. Within the Network Dataset, the Service Area function [61] was used to generate travel time zones based on the location of pharmacies. The Service Area was solved by specifying multiple travel times as cutoff values, with 5-minute intervals. Since the travel time was computed for one-way travel, the values for time zones were doubled to represent round-trip intervals of 10 minutes.

To determine the one-way delivery time for drones, we initially generated a Euclidean distance layer to calculate the distance from proposed drone hub sites, assuming direct flight paths towards the recipients. Subsequently, the estimated delivery time in minutes was computed as Eq 1:

$$\text{Travel Time (minutes)} = \left\lceil T_t + T_d + \left( \frac{Dist}{D_{mps}} \right) \right\rceil / 60 \tag{1}$$

In this equation, $T_t$ and $T_d$ represent the estimated takeoff and drop-off times, which were fixed at 35 seconds and 45 seconds, based on operational data provided by DroneUp. *Dist* refers to the straight-line (Euclidean) distance between

the drone hub and the patient's location, measured in meters. $D_{mps}$ is the drone's flying speed, expressed in meters per second; for this study, we used a value of 22.35 m/s (approximately 50 mph). Dividing the sum by 60 converts the total time from seconds to minutes, yielding the estimated one-way delivery time for each location. While the outcome of drone travel time is represented with continuous values, we reclassified the raster into 10-minute intervals to facilitate comparison with car travel zones. Once both travel zones were delineated, we aggregated the population layers (total and 60+) within each travel zone using the Zonal Statistics function. This approach allows for a more straightforward analysis of travel efficiency between drones and traditional personal vehicles, providing valuable insights for optimizing transportation strategies.

## 2.5 Assessing patient vulnerability

Patient age, vehicular travel time to the nearest pharmacy, and flood interruption were used to assess vulnerability. We used building footprint data to represent patient locations. To determine the age of potential patients, we utilized 2020 census block data (Table 2), which provides population counts across different age groups for both males and females. We aggregated the population of individuals aged 60 and older from both genders and calculated the percentage of this age group relative to the total population in each census block. These percentage values were then assigned to each building footprint based on its intersection with the corresponding census block. For personal vehicle travel time, we used the Closest Facility function to calculate the travel time from each building footprint to the nearest pharmacy. We employed the Network Database created from the personal vehicle travel zone analysis as the travel model and imported the building footprint and pharmacy locations as incidents and facilities, respectively, for the analysis. The result provided the shortest driving routes along the road network, measured in minutes. We then joined the driving time data with the building footprint data using the unique incident identification numbers.

To assess the impact of floodwaters on the study area, we used 100-year flood zone data from the Federal Emergency Management Agency (FEMA) along with road network layers to evaluate the effect of flooding on properties. The Closest Facility function was employed, with pharmacy locations as the facilities, building footprints as the incidents, and flood zone layer as polygon barriers. With the analysis outcome, we categorized the impact conditions into four categories: not affected, detoured, blocked, and inundated (Fig 4).

After computing and compiling the three criteria for each building footprint, we calculated the vulnerability index, as summarized in Table 4. Equal intervals were used to assign values ranging from 1 to 5 for the percentage of the population aged 60 and older, as well as travel time in minutes. For floodwater interruption, values were assigned from 1 to 4 based on the level of interruption. We created two separate vulnerability index calculations: [1] one considering the percentage of the population aged 60 and older and personal vehicle travel time, referred to as VAT, and [2] another incorporating the percentage of the 60+age group, travel time, and floodwater interruption, referred to as VATF. This dual approach distinguishes between baseline vulnerability under current conditions (VAT) and the added impact of potential flooding (VATF), which highlight everyday travel challenges versus those heightened by flood events, respectively. Although the probability of a 100-year flood is 1% annually, this risk may increase due to climate change-induced sea-level rise [62].

For both VAT and VATF, higher values indicate greater vulnerability: VAT ranges from 2 to 10, while VATF ranges from 3 to 14. After computing both indices, we performed a Getis-Ord $G_i^*$ analysis [63] using the Hot Spot Analysis tool to determine if the spatial distribution of high and low vulnerability showed statistically significant concentrations. The $G_i^*$ was calculated as Eq 2:

$$G_i^* = \frac{\sum_{j=1}^{n} w_{i,j}x_j - \overline{X}\sum_{j=1}^{n} w_{i,j}}{S\sqrt{\frac{\left[n\sum_{j=1}^{n} w_{i,j}^2 - \left(\sum_{j=1}^{n} w_{i,j}\right)^2\right]}{n-1}}}$$

(2)

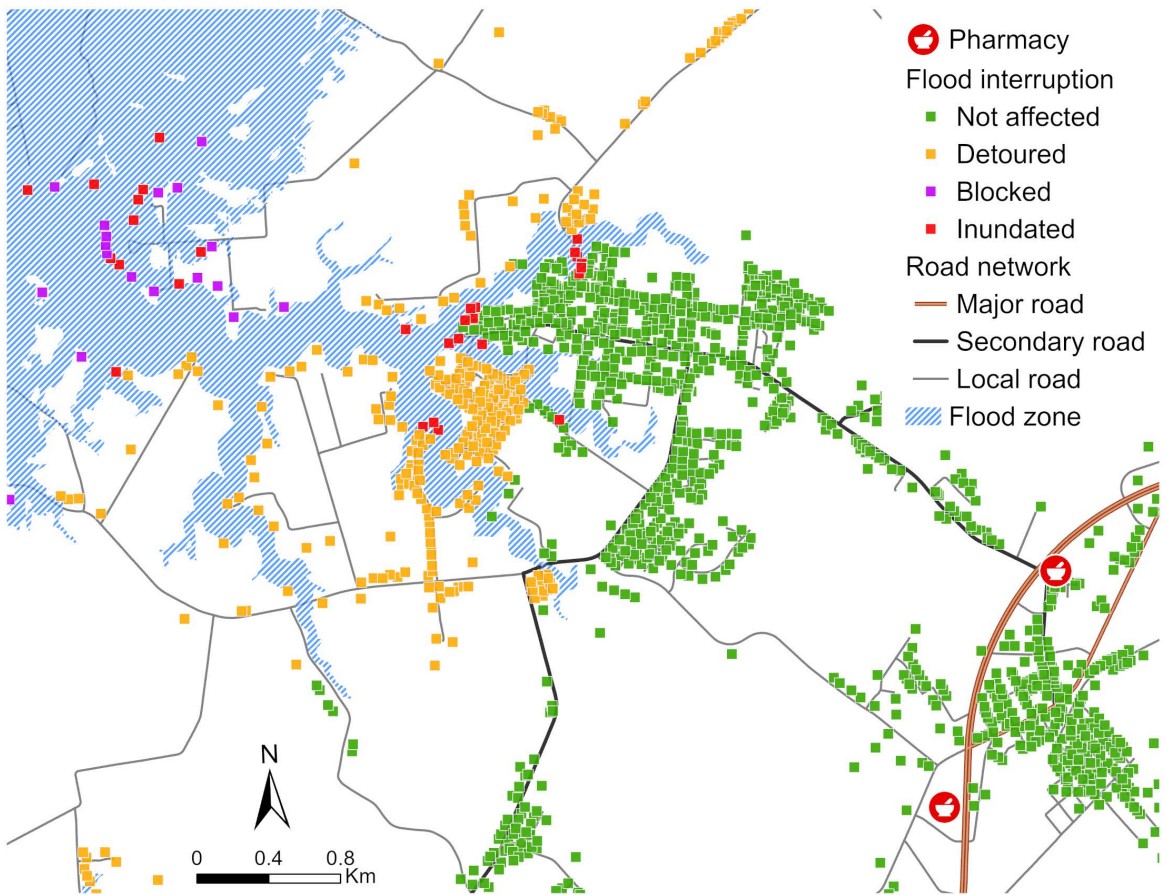

**Fig 4. A map illustrating the impact of floodwater on patients' vehicular transportation to their nearest pharmacy.**

**Table 4. Scoring criteria for patient vulnerability calculation.**

| Percentage of population age 60+ | Assigned value | Personal vehicle travel time (Round Trip, Minutes) | Assigned value | Flood interruption condition | Assigned value |
|---|---|---|---|---|---|
| ≤ 20% | 1 | ≤ 10 mins | 1 | Not affected | 1 |
| 21–40% | 2 | 11–20 mins | 2 | Detoured by floodwater | 2 |
| 41–60% | 3 | 21–30 mins | 3 | Blocked by floodwater | 3 |
| 61–80% | 4 | 31–40 mins | 4 | Inundated by floodwater | 4 |
| > 80% | 5 | > 40 mins | 5 | | |

In this equation:

- $x_j$ represents the vulnerability index value for building footprint $j$, capturing its relative vulnerability based on the VAT or VATF index.

- $w_{i,j}$ is the spatial weight between building footprints $i$ and $j$, indicating the degree of spatial proximity between the two locations, as defined by the spatial weights matrix.

- $n$ is the total number of building footprints included in the analysis.

$\overline{X}$ • is the mean vulnerability index value for all building footprints in the study area, calculated using Eq. (3).

• $S$ is the standard deviation of the vulnerability index values across all building footprints, calculated using Eq. (4).

$$\overline{X} = \frac{\sum_{j=1}^{n} x_j}{n}$$

(3)

This equation computes the mean vulnerability index by summing the vulnerability values of all building footprints and dividing by the total number of building footprints $n$.

$$S = \sqrt{\frac{\sum_{j=1}^{n} x_j^2}{n} - \left(\overline{X}\right)^2}$$

(4)

This equation calculates the standard deviation by measuring the dispersion of vulnerability index values around the mean, providing a basis for standardizing the statistic in Eq. (2).

The outcome $G_i^*$ were treated as z-scores to determine statistically significant clusters of lower (cold spots) and higher (hot spots) vulnerability. For the search distance, we used the average distance between 30 neighboring building footprints, which was 345 meters.

Finally, since drone delivery is a nascent technology, patients may not fully embrace the service [30,64]. Prioritizing stations that serve the most vulnerable patients can be particularly beneficial when resources are limited. To prioritize drone stations, we calculated the Euclidean distance from each building footprint to the nearest drone station to better understand how vulnerability indices are distributed in relation to each station. By examining these outcomes, we can assess the proximity of high-vulnerability areas to drone stations and determine which stations should be prioritized for resource allocation and response efforts.

## 3. Results

### 3.1 Travel time zone and served population

Fig 5 maps the delineated travel time zones across the ES, emphasizing the magnitude of time savings possible with drone delivery as compared with vehicular pharmacy trips. The travel time zones for personal vehicles follow the existing road network patterns (Fig 5a), whereas the drone travel time zones expand in near-perfect circles radiating from each drone station (Fig 5b). As shown Fig 5a, patients in some remote communities face round-trip travel times of up to 50 minutes to reach the nearest pharmacy, often requiring navigation of rural roads that can be further affected by flooding or other access challenges. In contrast, Fig 5b demonstrates that more than 80% of patients on the ES can receive their medications via drone within 10 minutes, underscoring the efficiency and potential reach of this delivery method. Beyond reducing travel time, drone delivery provides a critical alternative for patients who lack reliable vehicular transportation or who face mobility or accessibility limitations. For instance, residents of Tangier Island currently rely on boats or planes for medication deliveries due to the absence of vehicle access to pharmacies. Establishing a drone hub on the island eliminates these barriers, enabling patients to receive medications safely and reliably in less than 10 minutes. However, due to no-fly zone restrictions, a portion of patients in the northeastern ES are not reachable by drone delivery services.

Using the population estimation tool outlined in Section 2, it is possible to quantify the benefits of drone delivery specific to older patients who are likely more vulnerable. As shown in Fig 6, over 82% of the total population and those aged 60 + can be reached by one-way drone delivery within 10 minutes, whereas less than 38% of the ES population can complete a round trip drive to a nearby pharmacy within the same time frame. Following the 10-minute travel zone, 31%, 24%, 7%, and 1% of the total population would require 20, 30, 40, and 50 minutes, respectively, to reach the nearest pharmacy

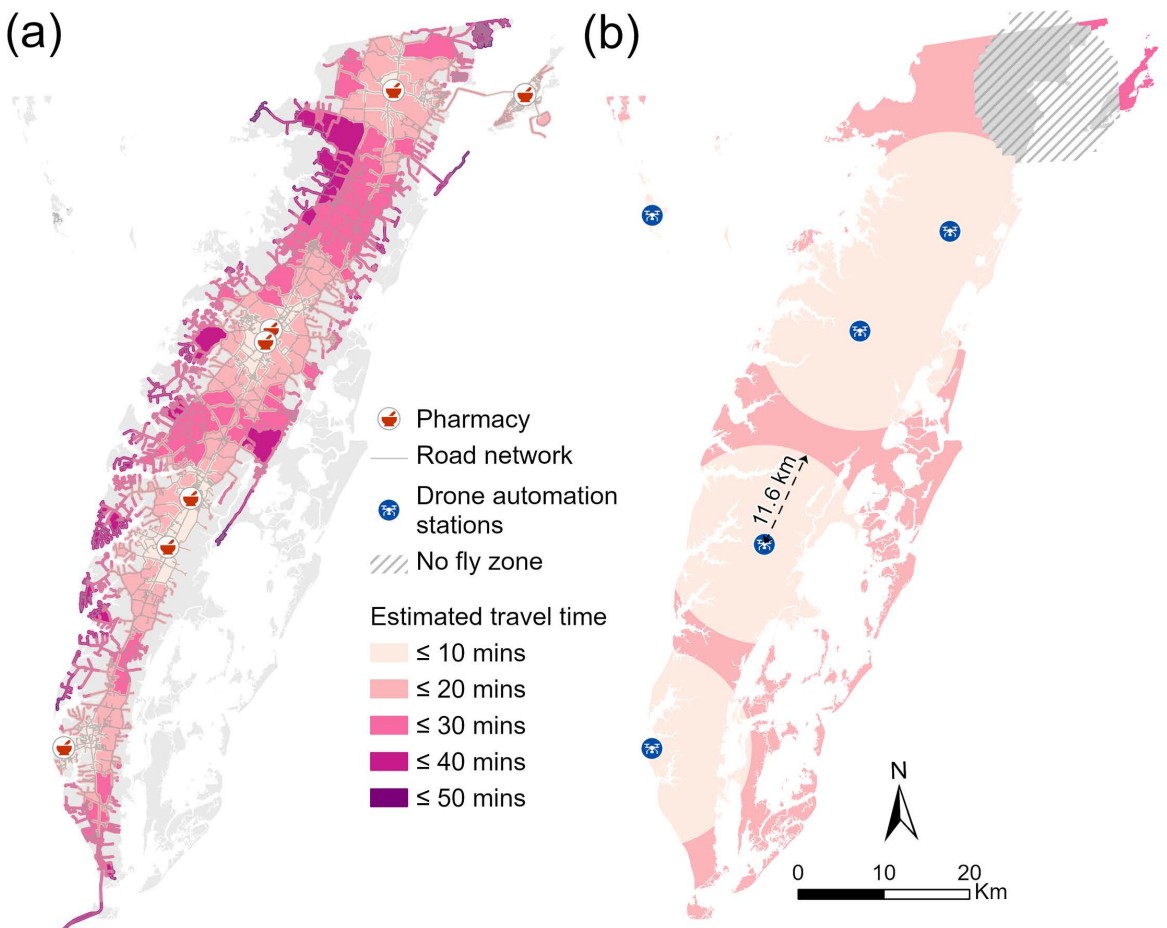

**Fig 5. Delineated travel time zones for round-trip personal vehicle driving (a) and one-way drone delivery (b) based on network analysis and Euclidean distance calculations.**

by personal vehicle, while 13% and 4% of the population can be served by drones within 20 and 30 minutes. The results for the population aged 60 and older followed a similar pattern, with differences of up to 3%. Overall, more than 99% of patients can be reached by drone within 30 minutes, whereas approximately 10% of those aged 60 and older would need over 30 minutes of driving time to access the nearest pharmacy. Note that residents of Tangier Island (438 individuals, including 180 aged 60 and older), were excluded from personal vehicle travel calculations because they are unable to drive to any pharmacies.

### 3.2 Patient vulnerability indices

A spatial analysis of the vulnerability indices introduced in Section 2.5 reveals distinct patterns of concentration and distribution across the ES, highlighting disparities between VAT and VATF outcomes. Fig 7 illustrates hot and cold spots identified by the two vulnerability indices, which measure patient vulnerability based on age, travel time, and flood exposure. Blue points indicate clusters of lower vulnerability values at the 99% confidence level, while red points represent clusters of higher vulnerability, highlighting locations where patients face greater challenges accessing pharmacies. The $G_i^*$ analysis for VAT (Fig 7a), which considers only the percentage of the population aged 60 and older and travel times,

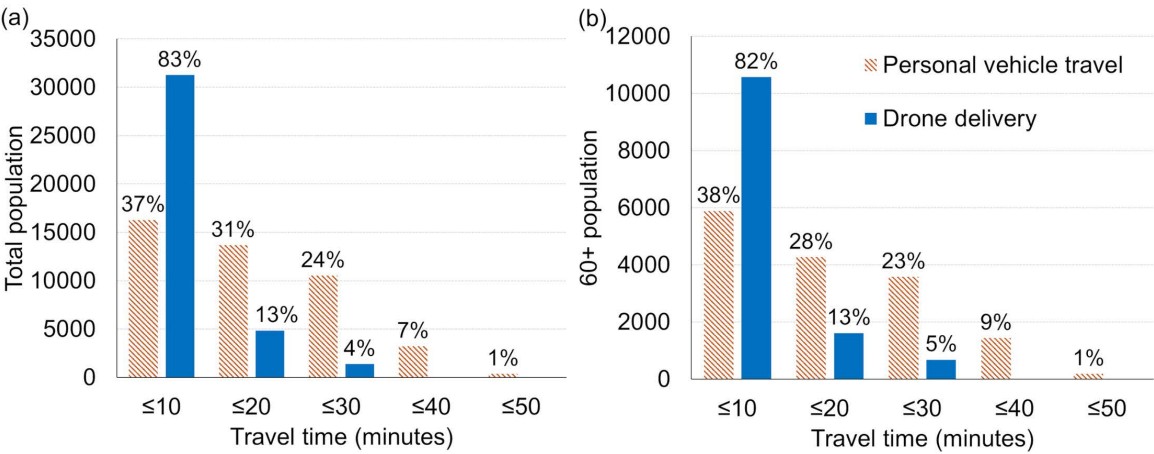

**Fig 6. Estimated total population (a) and population aged 60 and older (b) within travel time zones for round-trip personal vehicle travel to nearby pharmacies and one-way drone delivery.**

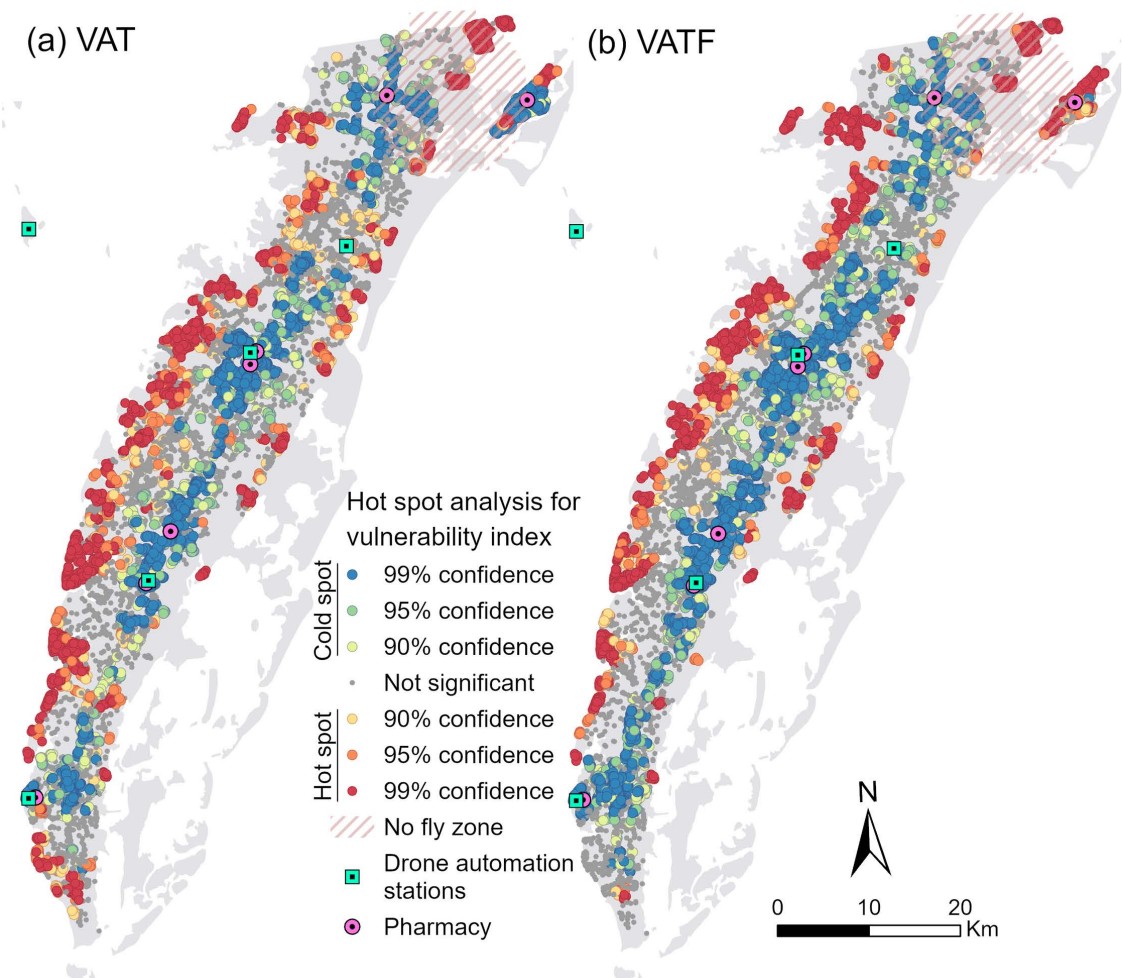

**Fig 7. Results of Getis-Ord G$_i^*$ spatial distribution analysis for the two vulnerability indices: (a) VAT and (b) VATF, with blue indicating low (cold spots) and red indicating high (hot spots) vulnerability clusters.**

shows greater spatial heterogeneity compared to the VATF results (Fig 7b). In contrast, the VATF results display less varied spatial patterns of hot and cold spots, largely influenced by relative elevation. The VATF outcome highlights a higher concentration of hot spots along the western coast of the ES, driven by elevated vulnerability values assigned to low-lying, flood-prone areas. Cold spots, corresponding to areas of lower vulnerability, are mainly located in the central part of the ES along U.S. Route 13 (Fig 1b), where elevation is comparatively higher.

The difference between the VAT and VATF outcomes is particularly pronounced in the Chincoteague Island area in the northeast. Under the VAT results, lower vulnerability values are concentrated near the island's pharmacy, with higher values observed among patients located farther away, particularly in areas with a higher proportion of older residents. In contrast, the VATF analysis indicates that most of the island exhibits elevated vulnerability, driven by its low-lying, flood-prone terrain. Only a few localized areas with relatively higher elevation and a smaller percentage of residents aged 60 and older show lower VATF values.

The spatial distribution of VAT and VATF indices varies notably among the four nearest drone automation stations, underscoring differences in service reach and vulnerability patterns across the ES. For clarity, the stations are numbered 1–4 from north to south in Table 5. The two northern stations cover a broader service area, reaching a greater number of residences, while the southern stations, particularly Station 4, serve fewer patients due to a lower density of residences within their flight range (Table 5). Station 1 exhibits the highest mean values for both VAT and VATF indices, indicating that it serves a population with greater baseline vulnerability as well as heightened vulnerability under flood conditions. For the VAT index, Stations 1 and 3 stand out with higher mean values (4.64 and 4.50); however, Station 3 displays a wider interquartile range and a larger number of high-scoring patients, suggesting greater variability in vulnerability levels (Fig 8). For the VATF index, Station 1 not only has the highest mean and median values but also a wider interquartile range, reflecting the compounding effects of flood risk in its service area. In contrast, Station 4, the southernmost station, serves the smallest population base and shows the lowest mean values for both indices, indicating that patients in its coverage area are comparatively less vulnerable under both baseline and flood-affected scenarios.

## 4. Discussion

### 4.1 Drone delivery efficiency and healthcare vulnerability hot spots

Our findings demonstrate that drone-based medical delivery significantly reduces travel time for rural residents on the ES, with the greatest benefits for vulnerable populations. While personal vehicle trips to pharmacies can take up to 50 minutes, over 80% of ES patients can receive drone deliveries within 10 minutes, including those on Tangier Island, where traditional transportation is unavailable. Spatial analysis of vulnerability indices reveals concentrated high-risk areas along the western coast, influenced by low elevation and flood susceptibility. Drone service reach varies across automation stations, with northern stations covering more patients and higher vulnerability scores. These findings highlight the potential of drones to enhance equitable healthcare access in flood-prone rural communities.

**Table 5. Number of building footprints, along with the mean and standard deviation of VAT and VATF indices, calculated for each nearest drone automation station. Note that building footprints within no-fly zones were excluded from this analysis.**

| Drone Automation Station | Building Footprint Count | VAT (value range: 2–10) | | VATF (value range: 3–14) | |
|---|---|---|---|---|---|
| | | Mean | Standard deviation | Mean | Standard deviation |
| 1. Riverside Shore Medical Center at Metompkin | 5883 | 4.64 | 1.43 | 6.61 | 2.30 |
| 2. Riverside Shore Memorial Hospital | 6211 | 4.31 | 1.44 | 5.99 | 2.25 |
| 3. Riverside Eastern Shore Family Medicine | 5045 | 4.50 | 1.71 | 5.89 | 2.09 |
| 4. Riverside Cape Charles Medical Center | 3232 | 4.34 | 1.36 | 5.41 | 1.48 |

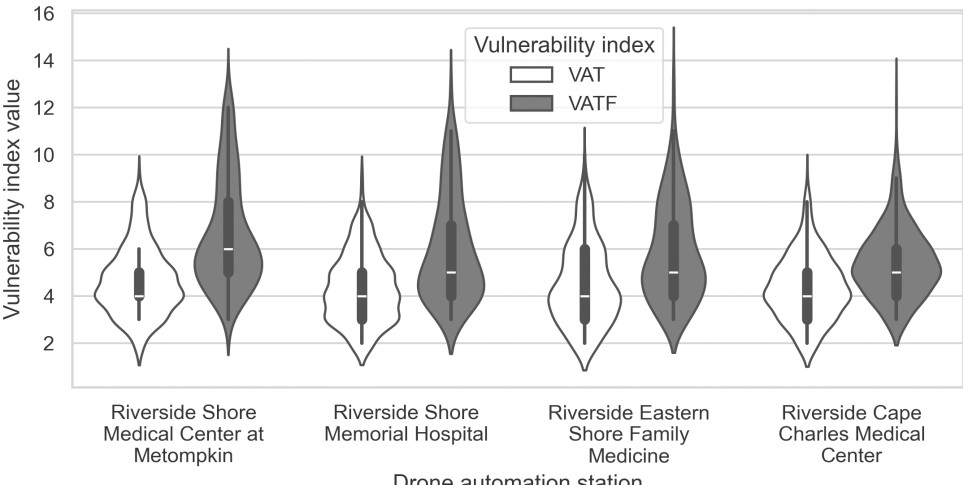

**Fig 8. Violin plots of VAT and VATF indices for drone stations arranged from north to south.** White bars represent median values, with thicker lines indicating the interquartile range. The plots show the kernel density estimation boundaries, where wider curves correspond to higher numbers of patients.

The travel zone analysis demonstrates that drone delivery provides significantly faster service than patient travel by personal vehicle, especially for residents of remote or waterfront communities (Fig 5). Unlike ground transportation, drones fly directly to their destinations at consistent speeds, avoiding traffic delays and road speed limits. With the exception of the Chincoteague area, where delivery may take up to 30 minutes due to no-fly zone restrictions, the majority of the ES can be reached by drone within 20 minutes. In contrast, only two pharmacies are located in two populated coastal towns (i.e., Chincoteague and Cape Charles), while the remaining five are concentrated along the main corridor of the ES (Fig 2b). As a result, patients in outlying areas must drive longer distances using slower secondary or tertiary roads, often requiring 40–50 minutes to reach a pharmacy by personal vehicle.

Along the ES, drones have the potential to significantly improve medication delivery times, particularly for older adults and residents of remote areas. Within 10 minutes, drones can reach over 80% of both the total population and those aged 60 and older. In contrast, fewer than 40% of either group can access a pharmacy within the same timeframe using round-trip personal vehicle travel (Fig 6). While overall travel times by car are comparable between age groups, a slightly higher proportion of the older population (~10%) falls within zones requiring more than 40 minutes of round-trip travel, compared to ~8% of the total population. This suggests that older adults are somewhat more concentrated in remote areas with limited access to nearby pharmacies. These individuals would particularly benefit from drone delivery services, which can offer substantially reduced travel times.

On Tangier Island, where land subsidence and erosion pose significant threats to the community [65], drone delivery may also be transformative. With a population of 438, of which approximately 41% (180 individuals) are aged 60 and older, Tangier Island has only one grocery store and no pharmacy. Currently, most medical services are provided by a single healthcare center, with supplies delivered by a single-engine light airplane [66,67]. While emergency medical support is available via helicopter from the Maryland State Police [66], establishing a drone delivery service and on-site drone station could substantially enhance medical supply access for this remote community.

The hot spot analysis of the VAT and VATF indices highlights the critical roles of travel time, elevation, and flood risk in determining service effectiveness. While the overall hot spot patterns for both VAT and VATF are similar, the VAT index, which accounts only for age and travel time, shows that older patients with longer travel distances face heightened

vulnerability. In contrast, the VATF analysis more distinctly highlights concentrated areas of high vulnerability in remote, waterfront communities, primarily due to their low-lying elevations (Fig 7). This pattern is particularly evident along the western coast of the Eastern Shore, where a series of isolated communities are accessible only by secondary roads extending toward the Chesapeake Bay (Fig 1). Notably, although both Cape Charles and Chincoteague are waterfront communities with pharmacies, only Chincoteague exhibits VATF hot spots. This distinction is explained by Chincoteague's lower elevation, where 75% of building footprints are inundated during a 100-year flood event, significantly limiting access to its local pharmacy.

In the context of service prioritization, spatial analysis using the VAT and VATF indices suggests that Station 1 in the northern ES should be prioritized for drone-based medication delivery. Under the VAT index, which considers age and travel time, Stations 1 and 3 show higher mean vulnerability scores, indicating their importance in serving older patients with limited access. However, when flood risk is incorporated into the VATF index, Station 1 emerges as the most critical hub, with the highest mean VATF score and the largest number of patients within its coverage area (Table 5 & Fig 8). This is largely due to its geographic distance from existing pharmacies—most other stations are located near pharmacies, leading to reduced vulnerability in their surrounding areas. Additionally, Station 1 covers the remote Chincoteague area, where a combination of physical isolation, limited infrastructure, and flood susceptibility exacerbates access challenges. These findings underscore the need to account for geographic complexity and environmental risk in the strategic planning of drone station placement, particularly in rural and flood-prone regions.

## 4.2 Scalable applications and policy implications

The geospatial analysis framework developed in this study is designed to be scalable and generalizable to other regions. By relying exclusively on publicly available datasets from federal, state, and county sources (Table 2), the methodology protects patient privacy while ensuring broad applicability. Although parcel data was obtained locally to target residential land use, most other datasets, such as road networks, building footprints, and elevation, are widely accessible across the U.S. This allows for straightforward replication in other medically underserved or flood-prone communities, making the approach a flexible and resource-efficient tool for planning drone-based healthcare delivery in diverse rural settings.

While some indices have been proposed for identifying health risks at a national level, such as Social Vulnerability Index used by the CDC and the Agency for Toxic Substances and Disease Registry (ATSDR) [68–70], the indices developed in this study operate at a finer, individual level, offering a more precise tool for evaluating delivery strategies. This approach can be adapted to other rural areas in Virginia or regions with similar geospatial data availability. By replicating this process, local and regional authorities can more effectively identify vulnerable hot spots, optimize healthcare resource allocation, and support the long-term viability of rural communities.

This study demonstrates feasible and scalable benefits of drone delivery systems in rural environments. As discussed in Section 2.3, future technological and regulatory developments may permit extended drone flying hours and operations in a broader range of weather conditions [58]. To engage the community and foster education, Riverside Health surveyed ES residents about their perceptions and experiences with drone delivery both before and after the medical delivery project, fostering awareness and support. Additionally, several residents were hired and trained by DroneUp as drone operators and operations personnel, promoting economic engagement and local expertise [58]. Employing and training residents as drone operators fosters economic opportunities and community engagement, enhancing program sustainability. This scalable model offers a transformative approach to improving healthcare access and resilience in underserved communities.

## 4.3 Study limitations and future directions

While our analysis provides valuable insights, several limitations persist. Regarding the comparison of travel times of drones and personal vehicles. In constructing the network dataset for travel time calculations, road hierarchy and speed limits were considered. Additional factors, such as restrictions on turns, variable traffic conditions [71], and speed

enforcement [72], could enhance the precision of these estimates. Previous research has also indicated that drivers often exceed posted speed limits [73,74], with speed preferences influenced by factors such as age, normlessness scale, and self-assessed driving skills [75]. Moreover, the drone travel times were computed using a fixed flight speed and station locations. Future research could investigate more sophisticated drone delivery systems, such as models that utilize multiple drone sizes with varying speed limits (e.g., Raj and Murray [76]), networks that allow drone exchanges between stations (e.g., Cokyasar, Dong [77]), or the inclusion of loading times from suppliers to drone stations. Additionally, recent advancements in AI-enabled drone logistics have demonstrated the potential of intelligent routing and operational optimization to enhance delivery efficiency and scalability [78]. Incorporating such AI-driven approaches in future analyses could allow for dynamic route planning, adaptive scheduling, and optimized resource allocation, further strengthening the resilience and responsiveness of drone-based healthcare delivery systems in flood-prone rural regions.

While our analysis used fixed drone speeds and station locations to estimate delivery times, actual flight performance provides critical validation and operational context. In parallel with our modeling efforts, DroneUp conducted extensive test flights to assess the feasibility of drone-based medical deliveries on ES (Fig 2). A total of 352 actual and 93 simulated flights were completed using the Prism Sky (Fig 2b) and the Swoop Aero Kite. The Prism Sky averaged a delivery time of 5.1 minutes for distances between 1.8 and 2.5 miles, while the Swoop Aero Kite completed a 36.4-mile delivery in approximately 15 minutes. These tests help ground-truth assumptions about drone capabilities and delivery timing. To further support operational planning, an airspace awareness and routing analysis was conducted using historical flight track data to identify preferred corridors and avoid potential conflicts. Three patients prescribed antihypertensive medications participated in the drone delivery pilot and provided positive feedback. Additionally, a 15-mile test flight from Riverside Shore Memorial Hospital to Tangier Island demonstrated the feasibility of longer-range missions [79].

Another limitation is the presence of no-fly zones, in which 7,749 individuals, including 2,943 aged 60 and older, are unserved (Fig 7). Moreover, approximately 4% of the total population and 5% of the 60-plus population on the ES, specifically in Chincoteague, still require over 20 minutes for drone delivery due to longer distances and the need to navigate around no-fly zones. A viable alternative for these areas could be truck-based drone delivery (e.g., Yoo and Chankov [80], Yin, Li [81]). Future research could also explore comparative analyses between bulk vehicle delivery and drone delivery in rural settings (e.g., Chiang, Li [82], Kirschstein [83] to assess relative efficiency and cost-effectiveness. For example, Kirschstein [83] examined the energy demands of diesel trucks, electric trucks, and drones, finding that while electric trucks generally consume the least energy, drones are particularly competitive in rural contexts. Such comparisons could support the development of more sustainable and equitable delivery strategies to address healthcare access disparities in rural communities.

The scoring methodology used for VAT and VATF computation may underestimate the complexity of patient vulnerability. In this study, we employed an equal interval scoring system for the percentage of the population aged 60 and older, as well as for travel time, with assigned values ranging from 1 to 5. Additionally, flood interruption conditions were classified into four levels, corresponding to values from 1 to 4 (Table 4). However, this approach may oversimplify patient vulnerability. For instance, Ostchega, Fryar [55] reported hypertension prevalence rates of 22.4%, 54.5%, and 74.5% among adults aged 18–39, 40–59, and 60 and older, respectively, with notable variations by gender. Integrating these prevalence rates into the vulnerability calculation could enhance its accuracy. Moreover, the relationship between age and driving behavior is nuanced. While willingness to drive generally decreases with age and varies by gender [84,85], driving ability remains crucial for successful aging, as it is linked to independence and social interaction [86,87]. This underscores the need for a more sophisticated approach to vulnerability computation that accounts for health conditions and behavioral factors.

Flooding risk in this work was captured by the relatively low probability of a 100-year flood event, but chronic high-tide flooding presents a more urgent threat to coastal communities. For example, Hino, Belanger [88] examined the impact of high-tide flooding in Annapolis, Maryland, and found that visits to the historic downtown area decreased by 1.7%, resulting in local economic losses. According to National Oceanic and Atmospheric Administration [89], the tidal station

in Wachapreague (Fig 1b), located on the middle east of ES, predicts up to 17 days of high-tide flooding annually. With ongoing sea-level rise driven by climate change, the ES is expected to face even more severe flooding impacts in the future [42,90]. By 2050, high-tide flooding could occur on up to 65 days per year in Wachapreague [89].

Incorporating community-specific preferences and critical thresholds into vulnerability assessments can significantly enhance the effectiveness of drone-based healthcare delivery strategies. Future research may consider conducting a comprehensive community survey to identify key factors influencing patient preferences (e.g., Kim [64], Bafouni-Kotta, Villanueva [91]). For instance, Kimmel, Bono [92] utilized Medicaid claims data to explore the relationship between drive time and retention rates in HIV care. Their findings indicated that a 30-minute drive time significantly impacts retention in rural areas, while no such relationship was observed in urban settings. Identifying similar thresholds in rural healthcare access and incorporating these insights into vulnerability index computation could provide valuable guidance for strategic planning in drone-based service deployment. In addition, assessing human-drone interaction within local communities would be essential to understanding user perspectives, such as concerns about security, usability, and perceived usefulness, which could further refine and improve the implementation of drone services [93].

The geospatial dataset compiled in this study can also serve as a baseline for further analysis, including cost-benefit assessments of drone-based medicine delivery. The proposed methodology may aid stakeholders in rural communities in mobilizing healthcare resources to patients, rather than requiring patients to travel for healthcare services or medications [94]. For example, Haidari, Brown [31] found that drones saved approximately $0.08 per vaccine dose compared to land-based transportation in Mozambique. One rural community in British Columbia, Canada has used medical deliveries via drones to help support healthcare providers who might otherwise feel isolated and overburdened, and therefore more prone to leave [94]. Another community in rural West Texas has envisaged reducing the number of vehicle trips to transport supplies to rural clinics, especially since those transporting the supplies are often licensed medical professionals who might otherwise have spent time in the service of patients [95]. By combining these data-driven insights with local community needs, drone service providers may partner with healthcare organizations to transform the delivery of essential medical services, especially in remote, flood-prone areas. This would not only improve operational efficiency but also ensure timely access to critical healthcare, reducing the vulnerability of underserved rural populations.

## 5. Conclusion

This study evaluated the effectiveness of drone-based medication delivery in improving healthcare access for vulnerable rural populations, particularly aging residents in flood-prone coastal areas. Using the ES of Virginia as a case study, this study assessed healthcare accessibility in flood-prone coastal communities by comparing traditional vehicle-based pharmaceutical trips with drone-based deliveries, while also evaluating patient vulnerability. Travel times for personal vehicles were calculated using a hierarchical road network and round-trip travel assumptions, whereas drone delivery times were estimated using direct flight paths from drone stations. Population layers were then aggregated within travel zones to quantify access efficiency. To identify the most at-risk populations, two vulnerability indices were developed: VAT, incorporating patient age and vehicle travel time, and VATF, additionally accounting for floodwater interruptions. Hot spot analysis identified statistically significant clusters of high vulnerability, guiding the prioritization of drone stations for service deployment. Overall, this integrated approach demonstrated how drones can improve timely access to medications, particularly for the most vulnerable populations, and provided a scalable framework for optimizing resource allocation under both everyday and flood-impacted conditions.

The three main contributions of this study include: first, quantifying the substantial time savings of drone-based medication delivery compared to traditional vehicle travel; second, developing two spatial vulnerability indices that integrate demographic, geographic, and flood exposure data to identify and prioritize high-need patients; and third, evaluating drone hub performance and geographic reach under varying vulnerability profiles to guide effective placement and resource allocation. Together, these contributions demonstrate that drone delivery networks are not only feasible but also strategically

valuable, providing a more equitable and resilient alternative to vehicle-dependent healthcare access. This approach helps improve healthcare access by identifying high-need populations, optimizing drone hub placement, and providing a scalable framework for resilient, equitable medication delivery in underserved communities.

Future research may enhance travel time estimates by incorporating real-world driving behaviors, dynamic traffic conditions, and more flexible drone routing systems, including multi-drone networks, station exchanges, and variable drone speeds. The vulnerability scoring methodology may be improved by integrating health condition prevalence, behavioral factors, and community-specific thresholds could provide a more accurate assessment of patient needs. Additionally, future research may address service gaps in no-fly zones through alternatives like truck-based drone delivery, conducting community surveys, and assessing human-drone interactions are also recommended. Finally, the geospatial dataset developed here offers a valuable foundation for cost-benefit analyses and stakeholder collaboration to expand efficient, adaptive, and community-informed drone-based healthcare delivery in rural, flood-prone regions.

In summary, this study investigates the transformative potential of drone technology in enhancing healthcare access for vulnerable aging populations in rural, flood-prone areas. By integrating publicly available data to identify at-risk communities, the proposed tools not only address immediate healthcare delivery challenges but also contribute to broader public health initiatives. This innovative approach can serve as a model for other underserved regions, promoting resilience in healthcare systems and ensuring equitable access to essential services, ultimately improving health outcomes for marginalized populations.

## Acknowledgments

We appreciate the feedback and support provided by members of Riverside Health, DroneUp, Accomack Northampton Planning District Commission, Virginia Innovation Partnership Corporation, and Virginia Institute for Space Flight & Autonomy. We would also like to acknowledge Nathan Lam for his support and the reviewers and editors who improved our manuscript.

## Author contributions

**Conceptualization:** Yin-Hsuen Chen, Amro M. El-Adle, Kevin J. O'Brien, Heather G. Richter.

**Data curation:** Yin-Hsuen Chen.

**Formal analysis:** Yin-Hsuen Chen, Kevin J. O'Brien.

**Funding acquisition:** Heather G. Richter.

**Investigation:** Amro M. El-Adle, Kevin J. O'Brien, Taylor Wentworth.

**Methodology:** Yin-Hsuen Chen, Amro M. El-Adle, Kevin J. O'Brien.

**Project administration:** Kevin J. O'Brien, Heather G. Richter.

**Visualization:** Yin-Hsuen Chen.

**Writing – original draft:** Yin-Hsuen Chen, Amro M. El-Adle, Kevin J. O'Brien.

**Writing – review & editing:** Yin-Hsuen Chen, Amro M. El-Adle, Kevin J. O'Brien, Taylor Wentworth, Heather G. Richter.

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
