## [Decision Letter · Decision Letter 0]

21 May 2025

PONE-D-25-20755Drone-Based Medication Delivery for Flood-Prone Coastal Communities: Optimizing Access and EquityPLOS ONE

Dear Dr. Chen,

Thank you for submitting your manuscript to PLOS ONE. After careful consideration, we feel that it has merit but does not fully meet PLOS ONE’s publication criteria as it currently stands. Therefore, we invite you to submit a revised version of the manuscript that addresses the points raised during the review process.

We look forward to receiving your revised manuscript.

Kind regards,

Zeashan Hameed Khan, Ph.D.

Academic Editor

PLOS ONE

3. We note that Figure 2 in your submission contain copyrighted images. All PLOS content is published under the Creative Commons Attribution License (CC BY 4.0), which means that the manuscript, images, and Supporting Information files will be freely available online, and any third party is permitted to access, download, copy, distribute, and use these materials in any way, even commercially, with proper attribution. For more information, see our copyright guidelines: http://journals.plos.org/plosone/s/licenses-and-copyright.

4. We note that Figures 1,4,5 and 7 in your submission contain [map/satellite] images which may be copyrighted. All PLOS content is published under the Creative Commons Attribution License (CC BY 4.0), which means that the manuscript, images, and Supporting Information files will be freely available online, and any third party is permitted to access, download, copy, distribute, and use these materials in any way, even commercially, with proper attribution. For these reasons, we cannot publish previously copyrighted maps or satellite images created using proprietary data, such as Google software (Google Maps, Street View, and Earth). For more information, see our copyright guidelines: http://journals.plos.org/plosone/s/licenses-and-copyright.

a. You may seek permission from the original copyright holder of Figures 1,4,5 and 7  to publish the content specifically under the CC BY 4.0 license. 

Additional Editor Comments:

The paper requires serious effort to improve the contents and justify the novelty of their work. The authors must justify how and why their approach is better than existing works?

Reviewers' comments:

Reviewer's Responses to Questions

**Comments to the Author**

1. Is the manuscript technically sound, and do the data support the conclusions?

Reviewer #1: Partly

Reviewer #2: Partly

2. Has the statistical analysis been performed appropriately and rigorously? 

Reviewer #1: No

Reviewer #2: Yes

3. Have the authors made all data underlying the findings in their manuscript fully available?

Reviewer #1: No

Reviewer #2: No

4. Is the manuscript presented in an intelligible fashion and written in standard English?

Reviewer #1: Yes

Reviewer #2: Yes

5. Review Comments to the Author

Reviewer #1: This paper addresses the feasibility of using drone delivery for medicine compared to road transportation methods. The authors mostly emphasize on the reduced time as the most benefecial aspect.

During emergency situations, the reduced time is a crucial factor, but for general deliveries, maybe the role of reduced time is over emphasized. I think the study should have considered other crucial factors to make a solid case. Here are my observations:

1. Though delivery time is reduced for a single delivery in drones, traditional vehicles can deliver in bulk and can complete multiple deliveries at neighbouring region within a very short time thus significantly reducing overall time/delivery. Without accounting for this the comparison seems unfair. Other minor thing could be time to load from pharmacy to drone site shipment or similar issues.

2. There is no ballpark cost estimates/comparisons per delivery between the methods in the paper. While one driver can possibly deliver to many people in a day, a drone pilot might not be able to reach much people due to one shipment/flight requiring more skilled pilots. Other aspects like battery costs/replacement costs could challenge the financial feasibility as well. At least a rough estimate of costs/delivery would be helpful.

3. This report is based on a particular location. How generalizable is this study to other places? Some discussions would be helpful.

I appreciate the authors for addressing a problem which might benefit several people get access to healthcare. I would appreciate more if case for drone based delivery is more convincing.

Reviewer #2: 1. The abstract lacks quantitative evidence on how much drone delivery improves accessibility or reduces delays compared to traditional methods, which weakens the impact of its findings.

2. The proposed vulnerability indices are mentioned, but the abstract does not explain how they were validated or their effectiveness in prioritizing patients, leaving their practical utility unclear.

3. The introduction should clearly conclude with a distinct paragraph that highlights the novel contributions of your work.

4. The literature review should benefit from more explorations of previous studies.

5. The discussion section needs to be expanded to more thoroughly analyze the results.

6. The first paragraph of the conclusion should succinctly summarize the contributions of the study in past tense.

7. The second paragraph of the conclusion should provide clear and actionable future recommendations.

8. Some equations are not properly cited.

9. Please place the figures within the text for the next round of revision.

6. PLOS authors have the option to publish the peer review history of their article (what does this mean? ). If published, this will include your full peer review and any attached files.

**Do you want your identity to be public for this peer review?** For information about this choice, including consent withdrawal, please see our Privacy Policy .

Reviewer #1: No

Reviewer #2: **Yes: ** Luttfi A. Al-Haddad

---

## [Author Response · Author response to Decision Letter 1]

31 Jul 2025

Reviewer #1:

1. This paper addresses the feasibility of using drone delivery for medicine compared to road transportation methods. The authors mostly emphasize on the reduced time as the most beneficial aspect.

Response: Thank you for your comment. We agree that reduced travel time is a key advantage of drone delivery over personal vehicle travel, and this was a primary focus of our GIS-based analysis. In the discussion section (4.1), we also expanded on additional benefits, particularly for elderly residents and their caregivers—highlighting the convenience of avoiding long, often burdensome trips from remote areas to pharmacies typically located in larger, more centralized communities along major roads.

2. During emergency situations, the reduced time is a crucial factor, but for general deliveries, maybe the role of reduced time is over emphasized. I think the study should have considered other crucial factors to make a solid case. Here are my observations:

Response: We have revised the manuscript accordingly. Please see our detailed responses below.

3. Though delivery time is reduced for a single delivery in drones, traditional vehicles can deliver in bulk and can complete multiple deliveries at neighboring region within a very short time thus significantly reducing overall time/delivery. Without accounting for this the comparison seems unfair. Other minor thing could be time to load from pharmacy to drone site shipment or similar issues.

Response: We agree that bulk deliveries via traditional vehicles can be an effective service model for rural communities. Bulk deliveries, however, may not serve all customers on the Eastern Shore. For example, residents of Tangier Island (included in the study area) are not connected to the mainland by any bridge or tunnel. Therefore, the only delivery methods currently available in that area are ferries and small airplanes. Our current analysis focuses on individual patients using personal vehicles to retrieve their prescriptions, rather than on pharmacy-led bulk delivery. As such, we did not include a direct comparison between bulk vehicle delivery and drone delivery.

To clarify this scope, we have revised the manuscript by specifying “personal vehicle travel” throughout the text. Additionally, we have added a new paragraph in the Introduction section highlighting the importance of personal vehicle use in rural areas, particularly for healthcare access. We believe these revisions help better define the study’s focus and provide readers with appropriate context for interpreting our findings. Please refer to: “Personal vehicles are essential for mobility and access to critical services in rural communities, particularly for older adults. A national study of individuals aged 65–79 found that rural residents were 7% more likely than their urban and suburban peers to emphasize the importance of driving (Strogatz et al., 2019), underscoring their reliance on personal transportation. In rural North Carolina, individuals with a driver's license made 2.29 times more healthcare visits, while those with access to family or friends for transportation had 1.58 times more visits (Arcury et al., 2006), highlighting how transportation directly affects healthcare access. Yet, despite this heavy reliance, older drivers in rural areas face disproportionate barriers due to chronic health conditions, physical impairments, and age-related declines in driving ability (Lyman et al., 2001; Krasniuk & Crizzle, 2023). Without viable transportation alternatives, these challenges severely limit rural seniors' access to care and independence.”

We also added several sentences in Section 4.3, Study Limitations and Future Directions, addressing the comparison between truck-based and drone delivery, as well as the consideration of loading times from suppliers:

“ … or the inclusion of loading times from suppliers to drone stations.”

“Future research could also explore comparative analyses between bulk vehicle delivery and drone delivery in rural settings (e.g., Chiang et al., 2019; Kirschstein, 2020) to assess relative efficiency and cost-effectiveness. For example, Kirschstein (2020) examined the energy demands of diesel trucks, electric trucks, and drones, finding that while electric trucks generally consume the least energy, drones are particularly competitive in rural contexts. Such comparisons could support the development of more sustainable and equitable delivery strategies to address healthcare access disparities in rural communities.”

4. There is no ballpark cost estimates/comparisons per delivery between the methods in the paper. While one driver can possibly deliver to many people in a day, a drone pilot might not be able to reach much people due to one shipment/flight requiring more skilled pilots. Other aspects like battery costs/replacement costs could challenge the financial feasibility as well. At least a rough estimate of costs/delivery would be helpful.

Response: We appreciate the Referee’s suggestion. As noted in a previous response, this study specifically focuses on comparing personal vehicle travel with drone delivery, rather than bulk delivery by truck. We have updated Introduction section, Page 6 to address the concern of delivery cost as follows:

1. As an estimate, a recent consulting report by PricewaterhouseCoopers estimated that the average cost of drone delivery per item was between $6 to $25. The cost is expected to drop to $2 per delivery by 2034 (Gajewska et al., 2024).

2. In the United States, Walmart offers drone delivery at certain locations for $19.99 per order (Walmart, 2024).

3. A drone delivery company called Manna that operates in Ireland has achieved a cost of $4 per order (Bogaisky, 2025).

It is important to note that a single drone pilot is currently permitted to operate three drones at the same time. The pilot observes the three flights remotely. In the future, the FAA may allow a single drone pilot to operate even more drones at the same time to help reduce the cost of deliveries.

5. This report is based on a particular location. How generalizable is this study to other places? Some discussions would be helpful.

Response: We have included a paragraph to address this comment. Please refer to section 4.2: “The geospatial analysis framework developed in this study is designed to be scalable and generalizable to other regions. By relying exclusively on publicly available datasets from federal, state, and county sources (Table 2), the methodology protects patient privacy while ensuring broad applicability. Although parcel data was obtained locally to target residential land use, most other datasets, such as road networks, building footprints, and elevation, are widely accessible across the U.S. This allows for straightforward replication in other medically underserved or flood-prone communities, making the approach a flexible and resource-efficient tool for planning drone-based healthcare delivery in diverse rural settings.”

6. I appreciate the authors for addressing a problem which might benefit several people get access to healthcare. I would appreciate more if case for drone based delivery is more convincing.

Thank you for your thoughtful comment. We agree that making a stronger case for drone-based delivery is essential to highlight its potential in addressing healthcare disparities in rural communities. In response, we have expanded the Introduction section to include additional context on the regulatory approval of drone-based medical delivery, as well as current cost estimates reported by major companies such as Walmart.

Furthermore, in Section 4.3, we have added a detailed paragraph discussing real-world test flights conducted in the study area by DroneUp. This includes over 350 actual and 90 simulated flights, demonstrating the operational feasibility and delivery speed of drones across various distance ranges. We also highlight feedback from patients who participated in the pilot program, which reflects strong community acceptance and the perceived usefulness of drone delivery services.

These additions aim to present a more convincing and evidence-based case for the viability of drone-enabled healthcare access.

Reviewer #2:

7. The abstract lacks quantitative evidence on how much drone delivery improves accessibility or reduces delays compared to traditional methods, which weakens the impact of its findings.

Response: Thank you for your comment, we have revised the description of abstract to include quantitative evidence. Please see the revised abstract: “Compared to traditional vehicle travel, drone delivery reduced trip times from up to 50 minutes to under 10 minutes for more than 80% of the population, including elderly patients.”

8. The proposed vulnerability indices are mentioned, but the abstract does not explain how they were validated or their effectiveness in prioritizing patients, leaving their practical utility unclear.

Response: We have added descriptions of the spatial analysis used to examine the clustering of vulnerable patients and the prioritized station. Please refer to the revised abstract “These indices were examined using Getis-Ord Gi* spatial analysis, which identified statistically significant clusters of high-need patients, particularly around the northernmost drone station. The results reveal that elderly residents in remote, low-lying areas are especially vulnerable to missed prescriptions due to both transportation barriers and flooding.”

9. The introduction should clearly conclude with a distinct paragraph that highlights the novel contributions of your work.

Response: We have revised the second last paragraph of introduction to highlight the novel contributions of our study. Please refer to the Introduction: “Using Virginia’s Eastern Shore (hereafter ES) as a case study, this study makes three key contributions to the field of healthcare accessibility and disaster-resilient delivery systems. First, it quantifies the substantial time savings of drone-based medication delivery over traditional vehicle travel in rural, flood-prone areas, providing one of the first direct comparisons in a non-emergency context. Second, it introduces two spatial vulnerability indices that integrate demographic, geographic, and flood exposure data to identify high-need patients, offering a replicable framework for prioritizing healthcare interventions. Third, it evaluates drone hub performance and reach across varying vulnerability profiles, guiding more effective placement and resource allocation under operational constraints. Together, these contributions demonstrate the feasibility and strategic value of drone delivery networks for enhancing medication access in underserved coastal communities and offer a scalable model for broader implementation in similarly challenged regions worldwide.”

10. The literature review should benefit from more explorations of previous studies.

Response: We have added a new paragraph highlighting the importance of personal vehicles in rural areas and the additional transportation challenges faced by older populations. We also incorporated the work of Tomio et al. (2010) to provide further evidence on how flooding disrupts access to medication, and cited a news article by Popper (2015) that documents the first government-approved medical drone delivery in Virginia. Please refer to the Introduction section.

11. The discussion section needs to be expanded to more thoroughly analyze the results.

Response: We have revised the discussion section to provide a more thorough analysis of the results, as requested. Please refer to Section 4.1 for the updated content. In response to this comment, the word count for this section has been expanded from 552 to 831 words to offer deeper insights into the findings and their implications.

12. The first paragraph of the conclusion should succinctly summarize the contributions of the study in past tense.

Response: We have revised the conclusion to be more succinctly summarize the contribution in past tense. Please refer to the Conclusion section.

13. The second paragraph of the conclusion should provide clear and actionable future recommendations.

Response: We have included a new paragraph to provide clear and actionable future recommendations. Please see second paragraph in Conclusion section: “Future research should enhance travel time estimates by incorporating real-world driving behaviors, dynamic traffic conditions, and more flexible drone routing systems. Improving the vulnerability scoring methodology by integrating health condition prevalence, behavioral factors, and community-specific thresholds could offer a more accurate assessment of patient needs. Addressing service gaps in no-fly zones through alternatives like truck-based drone delivery and conducting community surveys to guide strategic planning are also recommended. Finally, the geospatial dataset developed here provides a valuable foundation for future cost-benefit analyses and stakeholder collaboration to expand drone-based healthcare delivery in rural, flood-prone regions.”

14. Some equations are not properly cited.

Response: We have revised the relevant sections to ensure that all equations are properly cited.

15. Please place the figures within the text for the next round of revision.

Response: We have inserted figures within the text.

---

## [Decision Letter · Decision Letter 1]

18 Aug 2025

PONE-D-25-20755R1Drone-Based Medication Delivery for Flood-Prone Coastal CommunitiesPLOS ONE

Dear Dr. Chen,

Thank you for submitting your manuscript to PLOS ONE. After careful consideration, we feel that it has merit but does not fully meet PLOS ONE’s publication criteria as it currently stands. Therefore, we invite you to submit a revised version of the manuscript that addresses the points raised during the review process.

We look forward to receiving your revised manuscript.

Kind regards,

Zeashan Hameed Khan, Ph.D.

Academic Editor

PLOS ONE

Journal Requirements:

**Additional Editor Comments:**

The revised version has significantly improved but still needs some minor corrections.

Reviewers' comments:

Reviewer's Responses to Questions

**Comments to the Author**

1. If the authors have adequately addressed your comments raised in a previous round of review and you feel that this manuscript is now acceptable for publication, you may indicate that here to bypass the “Comments to the Author” section, enter your conflict of interest statement in the “Confidential to Editor” section, and submit your "Accept" recommendation.

Reviewer #2: (No Response)

Reviewer #3: (No Response)

2. Is the manuscript technically sound, and do the data support the conclusions?

Reviewer #2: (No Response)

Reviewer #3: Yes

3. Has the statistical analysis been performed appropriately and rigorously? 

Reviewer #2: (No Response)

Reviewer #3: Yes

4. Have the authors made all data underlying the findings in their manuscript fully available?

Reviewer #2: (No Response)

Reviewer #3: Yes

5. Is the manuscript presented in an intelligible fashion and written in standard English?

Reviewer #2: (No Response)

Reviewer #3: Yes

6. Review Comments to the Author

Reviewer #2: Minor English proofreading is required, please look for typos as they do exist within the manuscript.

Reviewer #3: Paper Review and Comments

1. Title and Abstract

• Title: Drone-Based Medication Delivery for Flood-Prone Coastal Communities

• Abstract:

Access to healthcare remains a critical challenge for rural populations, particularly in flood-prone coastal communities where transportation barriers limit access to essential medical services. This study evaluates the effectiveness of drone-based medication delivery in improving healthcare accessibility for vulnerable populations on Virginia’s Eastern Shore. Compared to traditional personal vehicle travel, drone delivery reduced trip times from up to 50 minutes to under 10 minutes for more than 80% of the population, including elderly patients. Using publicly available datasets, we developed two transportation vulnerability indices that incorporate age, travel time, and flood risk to prioritize patients for drone-based pharmaceutical delivery. These indices were examined using Getis-Ord Gi* spatial analysis, which identified statistically significant clusters of high-need patients, particularly around the northernmost drone station. The results reveal that elderly residents in remote, low-lying areas are especially vulnerable to missed prescriptions due to both transportation barriers and flooding. Our approach demonstrates how drone delivery can reduce healthcare access disparities while offering a scalable and resilient framework for other medically underserved regions, especially under time or resource constraints.

2. Introduction

• Aim and Motivation:

o Aim and Motivation is stated in introduction part.

o However, it should be explained little more.

o More references and explanation should be added in introduction as the introduction is very less.

• Research Questions and Objectives:

o The research objectives are not mentioned in the paper, please mention the objectives of the paper.

• Literature Review:

o Some more related papers can be included in the related work section.

3. Methodology

• Clarity of Methods:

o The methods are stated clearly that is good.

• Innovativeness:

o The proposed approach is novel.

4. Results and Analysis

• Presentation of Data:

o Presentation of the data is good.

o Some more details and references can be added in the introduction areas.

o More details can be added in the other sections like methodology and conclusion sections.

• Analysis and Discussion:

o The results are well analyzed.

5. Conclusion and Contributions

• Summary of Findings:

o The conclusion and future scope are stated good.

o But state what innovative method can be used in the future.

o State how your approach can help in the relevant field?

• Contributions:

o The contributions to the field are stated nicely.

o Some more details can be added in the conclusion part.

6. Language and Writing Style

• Grammar and Clarity:

o Some very little mistakes in grammar, that can be revised.

7. References

• Relevance and Recency:

o The references are relevant.

o However the references provided are very less, some more references can added. I have also recommended some references you can add them and extend the referencing area.

• Formatting:

o References are in proper order.

o However some more related references can be added, some references are given below in the recommendations section.

8. Figures, Tables, and Equations

• Figures:

o The figures should be explain in more details inside the text.

• Tables:

o The tables should be explain in more details inside the text.

• Equations:

o Explain each parameter of the equation/algorithm.

9. Recommendations for Improvement

1. The introduction section can be furnished with some new papers like:.

a. https://doi.org/10.4108/airo.5855

b. Using the different AI methods : https://doi.org/10.1007/978-981-97-5979-8_8 , https://doi.org/10.1007/978-981-97-5979-8_7

other food deliravy robot: doi: 10.1109/ACCESS.2024.3355278.

c.

10. Please answer below question

How did drone delivery compare to traditional personal vehicle travel times for medication delivery in Virginia’s Eastern Shore?

What factors were incorporated into the transportation vulnerability indices developed in this study?

Which area was identified as having the highest concentration of vulnerable patients, and why?

What role did community engagement and collaboration with multiple organizations play in the project’s success?

What future improvements to the vulnerability scoring methodology and service delivery were recommended by the study?

Overall Evaluation

• Final Recommendation:

• paper format and presentation should modify based on template.

• Major Revisions.

7. PLOS authors have the option to publish the peer review history of their article (what does this mean? ). If published, this will include your full peer review and any attached files.

**Do you want your identity to be public for this peer review?** For information about this choice, including consent withdrawal, please see our Privacy Policy .

Reviewer #2: **Yes: ** Luttfi A. Al-Haddad

Reviewer #3: No

---

## [Author Response · Author response to Decision Letter 2]

4 Sep 2025

Response to Reviewers

Reviewer #2:

1. Minor English proofreading is required, please look for typos as they do exist within the manuscript.

Response: Thank you for your comment. We have reviewed the manuscript and corrected identified typos; please refer to the tracked changes document for details.

Reviewer #3:

2. Title and Abstract

• Title: Drone-Based Medication Delivery for Flood-Prone Coastal Communities

• Abstract:

Access to healthcare remains a critical challenge for rural populations, particularly in flood-prone coastal communities where transportation barriers limit access to essential medical services. This study evaluates the effectiveness of drone-based medication delivery in improving healthcare accessibility for vulnerable populations on Virginia’s Eastern Shore. Compared to traditional personal vehicle travel, drone delivery reduced trip times from up to 50 minutes to under 10 minutes for more than 80% of the population, including elderly patients. Using publicly available datasets, we developed two transportation vulnerability indices that incorporate age, travel time, and flood risk to prioritize patients for drone-based pharmaceutical delivery. These indices were examined using Getis-Ord Gi* spatial analysis, which identified statistically significant clusters of high-need patients, particularly around the northernmost drone station. The results reveal that elderly residents in remote, low-lying areas are especially vulnerable to missed prescriptions due to both transportation barriers and flooding. Our approach demonstrates how drone delivery can reduce healthcare access disparities while offering a scalable and resilient framework for other medically underserved regions, especially under time or resource constraints.

Introduction

• Aim and Motivation:

o Aim and Motivation is stated in introduction part.

o However, it should be explained little more.

o More references and explanation should be added in introduction as the introduction is very less.

Response: We appreciate the reviewer’s comment. While the original introduction was intentionally concise to maintain focus, we have expanded it by adding 11 additional references and contextual details to strengthen the background and motivation for our study. Because the reviewer did not specify the missing literature, we incorporated studies across multiple dimensions to enrich the introduction, including:

• Literature supporting research motivation

o Casey et al. (2002): financial constraints affecting rural pharmacy services

o Berenbrok et al. (2022); Law et al. (2013); Todd et al. (2015); Sharareh et al. (2024): disparities in access to pharmaceutical services between urban and rural communities

o Tharumia Jagadeesan & Wirtz (2021): review of studies on pharmacy accessibility

o Ranković Plazinić & Jović (2018): mobility limitations of elderly populations

o Wassmer et al. (2025): impact of flooding on healthcare accessibility

• Literature on medical delivery using drones

o Snouffer (2022): pilot programs for drone-based medical deliveries

• Literature highlighting existing gaps

o Comi & Savchenko (2021); Garus et al. (2024): comparisons of delivery methods

o Berenbrok et al. (2022); Sharareh et al. (2024): methodologies for calculating driving times

We believe these additions provide a stronger foundation for the aim and motivation of our study, clarifying the significance of addressing rural healthcare access challenges and the potential role of drone-based solutions.

3. Research Questions and Objectives:

o The research objectives are not mentioned in the paper, please mention the objectives of the paper.

Response: We have included the research objective in the last paragraph of the introduction. Please see “The primary research objective is to develop and assess a scalable framework that integrates drone-based delivery, spatial vulnerability analysis, and operational modeling to improve equitable healthcare access in disaster-prone rural regions.”

4. • Literature Review:

o Some more related papers can be included in the related work section.

Response: Thank you for your comments. We have added 11 additional references to the introduction section; details are provided in our response to Comment #2.

5. Methodology

• Clarity of Methods:

o The methods are stated clearly that is good.

• Innovativeness:

o The proposed approach is novel.

Response: Thank you for your comments.

6. Results and Analysis

• Presentation of Data:

o Presentation of the data is good.

o Some more details and references can be added in the introduction areas.

o More details can be added in the other sections like methodology and conclusion sections.

Response: Thank you for your comments. We have added 11 additional references to the introduction section to provide more context and support; details are provided in our response to Comment #2. We have added more details regarding the methodology in the introduction section: “this study compares traditional vehicle-based pharmaceutical delivery with drone-based systems in flood-prone coastal communities and evaluates patient vulnerability. Travel times were estimated for both delivery modes, and two vulnerability indices—one incorporating age and travel time (VAT) and another adding flood impacts (VATF)—were used to identify high-risk populations. Hotspot analysis guided prioritization of drone stations, demonstrating how drones can improve timely medication access for the most vulnerable residents.”

We have also added more details about methodology into the conclusion:“… this study assessed healthcare accessibility in flood-prone coastal communities by comparing traditional vehicle-based pharmaceutical trips with drone-based deliveries, while also evaluating patient vulnerability. Travel times for personal vehicles were calculated using a hierarchical road network and round-trip travel assumptions, whereas drone delivery times were estimated using direct flight paths from drone stations. Population layers were then aggregated within travel zones to quantify access efficiency. To identify the most at-risk populations, two vulnerability indices were developed: VAT, incorporating patient age and vehicle travel time, and VATF, additionally accounting for floodwater interruptions. Hotspot analysis identified statistically significant clusters of high vulnerability, guiding the prioritization of drone stations for service deployment. Overall, this integrated approach demonstrated how drones can improve timely access to medications, particularly for the most vulnerable populations, and provided a scalable framework for optimizing resource allocation under both everyday and flood-impacted conditions.”

7. • Analysis and Discussion:

o The results are well analyzed.

Response: Thank you for your comment.

8. Conclusion and Contributions

• Summary of Findings:

o The conclusion and future scope are stated good.

o But state what innovative method can be used in the future.

o State how your approach can help in the relevant field?

Response: In addition to a new paragraph on future directions in the Conclusion, we have edited the text to incorporate additional innovative methods: “Future research should enhance travel time estimates by incorporating real-world driving behaviors, dynamic traffic conditions, and more flexible drone routing systems, including multi-drone networks, station exchanges, and variable drone speeds. Improving the vulnerability scoring methodology by integrating health condition prevalence, behavioral factors, and community-specific thresholds could provide a more accurate assessment of patient needs. Addressing service gaps in no-fly zones through alternatives like truck-based drone delivery, conducting community surveys, and assessing human-drone interactions are also recommended. Finally, the geospatial dataset developed here offers a valuable foundation for cost-benefit analyses and stakeholder collaboration to expand efficient, adaptive, and community-informed drone-based healthcare delivery in rural, flood-prone regions.”

We have also added a statement regarding the contribution to the relevant field: “This approach helps improve healthcare access by identifying high-need populations, optimizing drone hub placement, and providing a scalable framework for resilient, equitable medication delivery in underserved communities.”

9. • Contributions:

o The contributions to the field are stated nicely.

o Some more details can be added in the conclusion part.

Response: We have included a paragraph regarding the main contributions of this manuscript in the conclusion section. Please refer to “The three main contributions of this study include: first, quantifying the substantial time savings of drone-based medication delivery compared to traditional vehicle travel; second, developing two spatial vulnerability indices that integrate demographic, geographic, and flood exposure data to identify and prioritize high-need patients; and third, evaluating drone hub performance and geographic reach under varying vulnerability profiles to guide effective placement and resource allocation. Together, these contributions demonstrate that drone delivery networks are not only feasible but also strategically valuable as more equitable and resilient alternatives to personal vehicle–dependent healthcare access in underserved coastal communities.”

10. Language and Writing Style

• Grammar and Clarity:

o Some very little mistakes in grammar, that can be revised.

Response: We have reviewed the manuscript and corrected identified typos; please refer to the tracked changes document for details.

11. References

• Relevance and Recency:

o The references are relevant.

o However the references provided are very less, some more references can added. I have also recommended some references you can add them and extend the referencing area.

Response: Thanks for your suggestion. The manuscript included 83 references in the prior round of revisions; we added 11 more references to the introduction to further strengthen the context, as detailed in our response to Comment #2. Additionally, we incorporated one of the specific references the reviewer suggested, as noted in our response to Comment #14. With these updates, the manuscript now cites 95 references, which we believe provides thorough and sufficient coverage.

12. • Formatting:

o References are in proper order.

o However some more related references can be added, some references are given below in the recommendations section.

Response: Thank you for your comment. We have added 11 more references to the introduction to further strengthen the context, as detailed in our response to Comment #2. We have included a reference that the reviewer suggested. Please see our response to comment #14.

13. Figures, Tables, and Equations

• Figures:

o The figures should be explain in more details inside the text.

• Tables:

o The tables should be explain in more details inside the text.

• Equations:

o Explain each parameter of the equation/algorithm.

Response: Thank you for your feedback. Since the comment did not specify which details were lacking, we focused on Figures 5–8 and Table 5 in the Results section to ensure their explanations are more comprehensive and better integrated. We hope this addresses your concern and improves the clarity and readability of the manuscript. We also have included more detailed explanations for equations and their parameters. Please refer to the Result section and the descriptions for equation.

14. Recommendations for Improvement

1. The introduction section can be furnished with some new papers like:.

a. https://doi.org/10.4108/airo.5855

b. Using the different AI methods : https://doi.org/10.1007/978-981-97-5979-8_8 , https://doi.org/10.1007/978-981-97-5979-8_7

other food deliravy robot: doi: 10.1109/ACCESS.2024.3355278.

Response: Thank you for providing these insightful articles. We appreciate the reviewer’s suggestions and have carefully considered each recommended reference. The paper “Moshayed et al., 2024 Robots in Agriculture: Revolutionizing Farming Practices” has been added to section 4.3 Study Limitations and Future Directions . This placement allows us to acknowledge its relevance to the broader field while keeping the introduction concise and closely aligned with the primary goals of our study.

We also reviewed the book chapters “Moshayedi et al., 2024 Meta-heuristic Algorithms as an Optimizer: Prospects and Challenges (Part I and II).” However, we found it challenging to establish a direct connection between meta-heuristic algorithms and our current application, which focuses on spatial and vulnerability-driven modeling rather than algorithm optimization.

Finally, regarding “Moshayedi et al., 2024 Design and Development of FOODIEBOT,” we determined that food delivery robotics fall outside the scope of this manuscript, which is centered on healthcare-focused drone delivery in flood-prone rural communities. For this reason, we did not include this reference.

15. Please answer below question

How did drone delivery compare to traditional personal vehicle travel times for medication delivery in Virginia’s Eastern Shore?

Response: Drone delivery was markedly faster. Over 80% of Eastern Shore patients could receive medications within 10 minutes by drone, whereas many remote communities faced vehicle round trips of 20-50 minutes to the nearest pharmacy. In places without practical vehicle access, such as Tangier Island, drones reduced delivery time to under 10 minutes and provided a viable alternative to boat or plane transport.

16. What factors were incorporated into the transportation vulnerability indices developed in this study?

Response: The vulnerability indices incorporated three main factors: patient age, vehicular travel time to the nearest pharmacy, and flood-related travel interruptions. Patient age was assessed by calculating the percentage of individuals aged 60 and older in each census block and assigning those values to building footprints. Vehicular travel time was determined by calculating the shortest driving routes from each building footprint to the nearest pharmacy. Flood-related interruptions were evaluated using FEMA 100-year flood zones and road networks to categorize each property’s access as not affected, detoured, blocked, or inundated. Two indices were created: VAT, which included age and travel time, and VATF, which added flood interruption to capture the additional impact of flooding on patient access.

17. Which area was identified as having the highest concentration of vulnerable patients, and why?

Response: The western coast of the Eastern Shore was identified as having the highest concentration of vulnerable patients. This area’s elevated vulnerability is largely due to its low-lying, flood-prone conditions, which, when factored into the VATF index, increased vulnerability scores and created concentrated hotspot patterns in the spatial analysis.

18. What role did community engagement and collaboration with multiple organizations play in the project’s success?

Response: Community engagement and collaboration were critical to the project’s success. Local stakeholders, healthcare providers, drone service experts, and academic researchers worked together to address complex transportation barriers and design an effective, data-driven drone delivery network. The Accomack-Northampton Planning District Commission facilitated community outreach, ensuring local needs were understood and addressed. Healthcare providers like Riverside Health guided the medical delivery requirements, while DroneUp provided the technical expertise and operations. Academic partners supported project management and data modeling. This transdisciplinary collaboration ensured that the system was both technically feasible and responsive to the community’s healthcare needs.

19. What future improvements to the vulnerability scoring methodology and service delivery were recommended by the study?

Response: We recommended refining the vulnerability scoring methodology by integrating additional factors, such as the prevalence of health conditions like hypertension and behavioral aspects related to driving patterns b

---

## [Decision Letter · Decision Letter 2]

17 Sep 2025

Drone-Based Medication Delivery for Rural, Flood-Prone Coastal Communities

PONE-D-25-20755R2

Dear Dr. Chen,

We’re pleased to inform you that your manuscript has been judged scientifically suitable for publication and will be formally accepted for publication once it meets all outstanding technical requirements.

Kind regards,

Zeashan Hameed Khan, Ph.D.

Academic Editor

PLOS ONE

Additional Editor Comments (optional):

The revised version is sufficiently improved and hence can be accepted in the present form.

Reviewers' comments:

Reviewer's Responses to Questions

**Comments to the Author**

1. If the authors have adequately addressed your comments raised in a previous round of review and you feel that this manuscript is now acceptable for publication, you may indicate that here to bypass the “Comments to the Author” section, enter your conflict of interest statement in the “Confidential to Editor” section, and submit your "Accept" recommendation.

Reviewer #2: (No Response)

Reviewer #3: All comments have been addressed

2. Is the manuscript technically sound, and do the data support the conclusions?

Reviewer #2: (No Response)

Reviewer #3: Yes

3. Has the statistical analysis been performed appropriately and rigorously? 

Reviewer #2: (No Response)

Reviewer #3: Yes

4. Have the authors made all data underlying the findings in their manuscript fully available?

Reviewer #2: (No Response)

Reviewer #3: Yes

5. Is the manuscript presented in an intelligible fashion and written in standard English?

Reviewer #2: (No Response)

Reviewer #3: Yes

6. Review Comments to the Author

Reviewer #2: (No Response)

Reviewer #3: The authors have carefully addressed all of my previous concerns, and the paper has undergone substantial revisions. I believe these changes have significantly improved the overall clarity, quality, and contribution of the work. The revised version is more coherent, and the presentation of results is much stronger compared to the earlier draft. Overall, I am satisfied with the responses provided and the improvements made.

7. PLOS authors have the option to publish the peer review history of their article (what does this mean? ). If published, this will include your full peer review and any attached files.

**Do you want your identity to be public for this peer review?** For information about this choice, including consent withdrawal, please see our Privacy Policy .

Reviewer #2: No

Reviewer #3: No

---

## [Editor Report · Acceptance letter]

PONE-D-25-20755R2

PLOS ONE

Dear Dr. Chen,

I'm pleased to inform you that your manuscript has been deemed suitable for publication in PLOS ONE. Congratulations! Your manuscript is now being handed over to our production team.

Kind regards,

on behalf of

Dr. Zeashan Hameed Khan

Academic Editor

PLOS ONE